# Compartmental inspiratory genioglossus electromyographic activity in supine, awake individuals with and without obstructive sleep apnoea

Lauriane Jugé[1,2] ⓘ, Peter G. R. Burke[1,3], Jade Yeung[1] ⓘ, Fiona Knapman[1] ⓘ, Elizabeth C. Brown[1,4], Alan Chiang[1] ⓘ, Danny J. Eckert[1,2,5] ⓘ, Jane E. Butler[1,2] ⓘ and Lynne E. Bilston[1,6]

[1]*Neuroscience Research Australia, Sydney, New South Wales, Australia*
[2]*Faculty of Medicine and Health, University of New South Wales, Sydney, New South Wales, Australia*
[3]*Macquarie Medical School, Faculty of Medicine and Health Sciences, Macquarie University, Sydney, New South Wales, Australia*
[4]*Prince of Wales Hospital, Sydney, New South Wales, Australia*
[5]*Adelaide Institute for Sleep Health, Flinders Health and Medical Research Institute, Flinders University, Adelaide, Australia*
[6]*Faculty of Engineering and IHealthE, University of New South Wales, Sydney, New South Wales, Australia*

Handling Editors: Harold Schultz & Ken O'Halloran

The peer review history is available in the Supporting Information section of this article (https://doi.org/10.1113/JP287943#support-information-section).

**Abstract figure legend** This study examined compartmental differences in multiunit EMG activity in the anterior and posterior sections of the horizontal and oblique genioglossus during quiet nasal breathing in awake, supine, non-obese individuals with and without OSA. Three key findings emerged: (1) phasic EMG was higher in the oblique than the horizontal compartments; (2) greater phasic activity in oblique compartments was linked to more negative nadir $P_{epi}$; and (3) OSA severity had minimal impact on genioglossus activation.

J. E. Butler and L. E. Bilston contributed equally as senior authors.
L. Jugé and P. G. R. Burke contributed equally as first authors.

The Journal of Physiology

**Abstract** Inspiratory-related genioglossus EMG activity is crucial to maintain upper airway patency. However, whether this activity differs between the oblique (middle) and horizontal (base) compartments or if they vary in people with obstructive sleep apnoea (OSA) is unknown. Here, intramuscular electrodes were inserted into the anterior and posterior regions of the horizontal and oblique genioglossal compartments in nine controls [apnoea–hypopnoea index (AHI) $\leq$ 5 events/h] and 45 OSA participants (AHI range 5–94.3 events/h). Multiunit EMG patterns were categorised as phasic (respiratory modulation) or tonic (no respiratory modulation) during nasal breathing in awake, supine participants. The effects of OSA status and genioglossus compartments were assessed through linear mixed models, controlling for nadir epiglottic pressure ($P_{epi}$) and repeated measures within participants. Phasic patterns occurred in 57.6% ($n = 106/184$) of compartments. Within phasic compartments, $\log_{10}$-transformed peak, phasic and tonic EMG (% maximum) were higher in the oblique than in the horizontal compartments. Additionally, more pronounced negative $P_{epi}$ correlated with increased $\log_{10}$-transformed phasic EMG in the anterior oblique (beta $= -0.075$, $P = 0.002$) and posterior oblique compartments (beta $= -0.080$, $P = 0.027$), but not in the horizontal compartments. Effects of OSA severity on activity patterns or EMG measurements were not significant. To conclude, the genioglossus exhibited regional (oblique–horizontal) variation in neural drive during awake inspiration. This compartmental activity appears to be driven by reflex activation in the oblique compartments, which increase phasic EMG. People with and without OSA have similar drive during wakefulness. Understanding the mechanisms driving efficient genioglossus dilatory activity is essential to develop targeted treatments for OSA that focus on pharyngeal muscle activity.

(Received 23 October 2024; accepted after revision 25 March 2025; first published online 14 April 2025)

**Corresponding author** Lynne E. Bilston: Neuroscience Research Australia, Margarete Ainsworth Building, Barker Street Randwick, NSW 2031, Australia.     Email: lynne.bilston@unsw.edu.au

## Key points

- Inspiratory genioglossus multiunit EMG activity is thought to vary across different neuromuscular compartments. However, it remains unclear whether obstructive sleep apnoea (OSA) affects this compartmental variability.
- During quiet nasal breathing in awake supine individuals, inspiratory genioglossus EMG normalised to maximum EMG was measured in four genioglossus neuromuscular compartments in individuals with and without OSA.
- Both tonic (no respiratory modulation, 42%) and phasic (respiratory modulation, 58%) activity patterns were observed during breathing.
- When a genioglossus compartment showed phasic activity, peak, phasic and tonic EMG activities were higher in the oblique than in the horizontal compartments. Furthermore, greater phasic activity was associated with a more negative nadir epiglottic pressure only in the oblique compartments.
- There was no additive effect of OSA severity on top of the more negative nadir epiglottic pressure, suggesting people with and without OSA received similar drive during inspiration during wakefulness.

**Lauriane Jugé** is a Senior Research Fellow in biomedical imaging at Neuroscience Research Australia (NeuRA) and conjoint Senior Lecturer at the Faculty of Medicine of the University of New South Wales (Sydney, Australia). She leads pioneering multidisciplinary research using multimodal imaging techniques to uncover novel mechanistic insights into physiological challenges, including sleep disorders and neural injury. Her expertise spans the spectrum from basic to clinical science. **Peter Burke** is a neurophysiologist by training and a Senior Lecturer at the Macquarie University (Sydney, Australia). His research focuses on how the mammalian brain regulates the upper airways, breathing and blood pressure in health and in obstructive sleep apnoea.

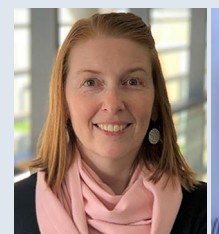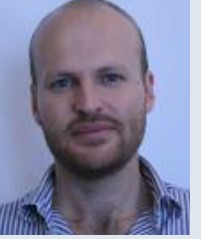

## Introduction

The genioglossus is the largest upper airway dilator muscle, playing a vital role in various tongue functions. These functions include maintaining airway patency during breathing and facilitating swallowing and speech. The muscle comprises two neuromuscular compartments (Mu & Sanders, 2010; Sanders & Mu, 2013). The *horizontal* compartment, located at the base of the genioglossus, consists of horizontal fibres that run posteriorly and are responsible for tongue protrusion by pulling the back of the oropharyngeal section of the tongue forward. The *oblique* compartment, in the middle of the genioglossus, contains oblique fibres that pull the nasopharyngeal section of the tongue downward and forward.

Based on morphological and histochemical studies, Mu and Sanders hypothesised that the horizontal compartment of the genioglossus may have a more prominent role than the oblique compartment in maintaining airway patency during inspiration. This is based on the observation that horizontal muscle fibre contractions may more effectively 'pull' the back of the tongue anteriorly than oblique muscle fibres, resulting in a larger widening of the oropharynx. Furthermore, a greater proportion of slow twitch (type I) fibres in the horizontal compartment, known for their endurance and ability to sustain prolonged activity, suggest that this compartment is better suited for the continuous and repetitive actions required for breathing than the oblique compartment (Mu & Sanders, 2000).

Our recent dynamic imaging work supports this (Jugé et al., 2020). When the tongue moves forward during inspiration, we observed greater anterior movement in the horizontal than in the oblique compartment during wakefulness (Cheng et al., 2008). This anterior movement captures the contraction of the tongue muscles, including genioglossus, during inspiration to dilate the airway (Cheng et al., 2014). This is followed by posterior movement during expiration as the muscles relax (Cheng et al., 2008). We observed this in people with and without obstructive sleep apnoea (OSA) (Brown et al., 2013; Jugé et al., 2020).

OSA is a common and serious sleep–breathing disorder characterised by repeated complete (apnoea) or partial (hypopnoea) collapses of the upper airway due to impaired upper airway function during sleep (Bilston & Gandevia, 2014; Edwards & White, 2011). A major contributor to upper airway collapse in OSA is inadequate neural drive to the genioglossus during sleep (Eckert et al., 2013; Horner et al., 1994). In awake people with OSA, inspiratory-related and tonic genioglossus activities are key to enlarging and/or stiffening the upper airway to protect it from collapsing during inspiration (Mezzanotte et al., 1992; Owens et al., 2012). Regional differences in genioglossus muscle activity have been reported in studies conducted during wakefulness. Luu et al. (2018) found that different types of genioglossus motor units were clustered, with inspiratory units primarily located in the posterior genioglossus and expiratory units primarily located near the border of the genioglossus and geniohyoid. Eastwood et al. (2003) and Vranish & Bailey (2015) found opposite anterior/posterior differences in genioglossus multiunit EMG during awake supine quiet nasal breathing in healthy adults. The former reported a trend for a greater tonic and phasic EMG anteriorly, and the latter observed significantly higher EMG in the posterior region. None of these studies compared EMG activation specifically in the horizontal and oblique compartments of the genioglossus during respiration or determined whether this activation is altered in individuals with OSA. Given that airway collapse most commonly occurs in the retropalatal region behind the soft palate (Boudewyns et al., 1997; Demin et al., 2002), understanding the fundamental processes that drive efficient dilatation of the oblique neuromuscular compartment and assessing whether it differs from observations within the horizontal compartment is crucial for the development of novel targeted treatment approaches such as neural stimulation or pharmaceutical interventions that modify muscle activity.

Thus, the two aims of this study were to determine during awake quiet breathing whether: (1) there are differences in multiunit EMG across oblique and horizontal genioglossus compartments and (2) this EMG activation is altered with OSA severity. We hypothesised that: (1) multiunit EMG activation would be greater in the genioglossus horizontal than in the oblique compartment, based on the previous histological and imaging works mentioned above; and (2) EMG activation would be greater in participants with higher OSA severity to compensate for the greater negative pharyngeal pressure generated during inspiration in OSA. Awake-state recordings are important for understanding the predisposition to airway collapse during sleep and may help to identify key neuromuscular features that differentiate OSA and non-OSA populations.

## Methods

### Ethical approval

The study was approved by the South Eastern Sydney Local Health District Human Research Ethics Committee [ref. 13/347 (HREC/13/POWH/745)]. It was conducted adhering to the 2013 *Declaration of Helsinki*, except for registration in a publicly accessible database. Participants gave informed written consent before commencement of the study.

## Participants

Sixty-three participants were recruited for the study. OSA severity was characterised by the apnoea–hypopnoea index (AHI) based on a standard overnight polysomnography study performed no more than 1 year prior to this study, except for three participants who had their polysomnography up to 2.4 years from the study but did not report body weight changes. All polysomnography data were scored according to the American Academy of Sleep Medicine v2.4 criteria (3% desaturation) (Berry et al., 2017). Potential participants with chronic illness or taking medication, including sedatives and psychoactive drugs, that could impact upper airway muscle activity or increase the risk of bleeding were excluded. People with a history of upper airway surgery were also excluded.

## Experimental setup, equipment and measurements

The experimental setup for the study is described in detail in Yeung et al. (2022). Briefly, four single-strand fine-wire hook monopolar electrodes [127 μm diameter stainless steel wire coated in Teflon (No. 791500 A-M Systems Inc., Carlsborg, WA, USA)] were inserted percutaneously into the genioglossus, targeting (1) the *anterior oblique*, (2) *posterior oblique*, (3) *anterior horizontal* and (4) *posterior horizontal* compartments following application of topical anaesthetic under the chin (Emla cream, 5% lignocaine and prilocaine; AstraZeneca, Australia) (Fig. 1). Electrodes targeting oblique and horizontal compartments were inserted to a depth of 1.5 and 0.5 cm above the superior margin of the geniohyoid, respectively. Anterior electrodes were positioned 1 cm posterior to the posterior edge of the mandibular symphysis, and posterior electrodes were positioned 2 cm further posterior. Ultrasound was used to visualise submental anatomy (7 MHz frequency,

iU22, Philips, Best, Netherlands). However, due to the lack of visible distinction between the oblique and horizontal compartments, after positioning, wires were assigned to one of the four tongue neuromuscular compartments based on the EMG pattern during swallowing: a single peak monophasic pattern in the horizontal compartment, and two smaller peaks biphasic response in the oblique compartments (Yeung et al., 2022). Each monopolar electrode had a 1.5 mm bared tip for recording and was referenced to a 20 mm surface electrode (Cleartrace™, Conmed Corporation, Utica, NY, USA) placed on the forehead. A ground electrode (12 × 13.2 cm, 3M™, Maplewood, MN, USA) was placed on the shoulder. EMG signals were band-pass filtered to between 30 and 1000 Hz. In addition, a pressure-tipped catheter (MPC500, Millar, Houston, TX, USA) was used to measure epiglottic pressure, with the catheter inserted through the nose and advanced until the tip is at the level of the epiglottis. Participants were also fitted with a nasal mask (ResMed, Sydney, Australia) connected to a pneumotachometer (700, Hans Rudolph Inc., Kansas City, MO, USA) and differential pressure transducer (DP15-16, Validyne Engineering Corp., Northridge, CA, USA) for measurement of airflow. All data were recorded using Spike2 software (v6.17 Cambridge Electronic Design, Cambridge, UK).

Participants were asked to perform tongue protrusions and 'dry' swallows as maximal manoeuvres, after a short period of being supine and lying awake. Specifically, for the tongue protrusions, participants were instructed to push the tip of the tongue as hard as possible against the back of their front teeth while being encouraged verbally. For the 'dry' swallows, they were also asked to swallow as hard and fast as possible. Each manoeuvre was repeated with 30 s between efforts. Following this, participants were asked to breathe quietly through their noses, during which recordings were obtained.

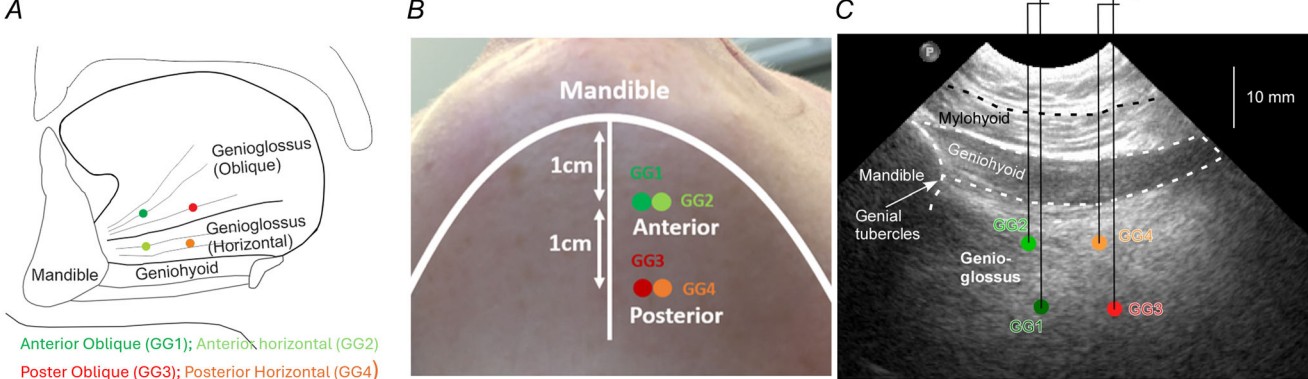

**Figure 1. Experimental setup**
Four submental fine wire electrodes (*A*, *B*) were inserted into an anterior and posterior location within the horizontal and oblique compartment of the genioglossus under ultrasound guidance (*C*). Figure adapted from Yeung et al. (2022). [Colour figure can be viewed at wileyonlinelibrary.com]

## Respiratory and muscle activity analysis

Nadir epiglottic pressure ($P_{epi}$), flow and EMG signals during periods of quiet nasal breathing were visually inspected, and periods of stable breathing and artefact-free signal recordings were selected for analysis.

**Respiratory parameters.** A modified version of a custom-designed, semi-automated script (Nguyen et al., 2017) was used to calculate respiratory parameters from the flow signal: inspiratory time ($T_i$), total breath duration ($T_{tot}$), tidal volume ($V_t$), minute ventilation ($V_i$), frequency of breathing ($F_b$) and mean inspiratory flow ($V_t/T_i$). These respiratory parameters were used to explore potential differences in breathing patterns between non-OSA and OSA participants, which may be influenced by variations in reflex pharyngeal muscle activation as a result of increased negative nadir $P_{epi}$ in OSA participants (Horner, 2012).

**Phasic and tonic EMG activity patterns.** Each recording was visually inspected by a single researcher (J.Y.) and assigned either as being 'phasic' or 'tonic' based on the pattern of activity in synchrony with the airflow signal during breathing. An example is shown below in Fig. 2, where phasic activity refers to signals with distinctive bursts of activity timed with breathing. In contrast, tonic signals show sustained background activity without any phasic activity (Belavý et al., 2009). Note that all wires with tonic pattern included in the analysis exhibited increased activity during tongue protrusions and swallowings.

**Multiunit EMG amplitude.** The four parameters of interest are shown in Fig. 3. (1) *Peak EMG* and (2) *Tonic EMG* amplitude, defined as the maximum and minimum rectified, moving time-averaged (100 ms) EMG, respectively, were calculated. (3) *Phasic EMG* amplitude was calculated as the difference between the peak EMG and nadir EMG. (4) *Central inspiratory EMG (hereafter Insp EMG)* was calculated as the EMG at the beginning of inspiration, identified by a sharply decreasing nadir $P_{epi}$ and the onset of inspiratory airflow. Insp EMG was quantified to provide insights into the neural control of breathing since central inspiratory drive from the respiratory pattern generator is initiated prior to airflow in the pharynx that additionally recruits reflex dilatory EMG (Butler, 2007; Strohl et al., 1980).

All rectified, moving time-averaged, EMG measurements were normalised to the maximum of the largest of at least two maximal tongue protrusions or swallows. EMG from non-suitable breaths that had indications of movement artefact, swallowing or strong single motor unit activity were excluded from the quantitative analysis.

## Statistical analysis

Data were analysed blinded to OSA status and severity. Participants with an AHI $\leq$ 5 events/h of sleep were classified as non-OSA participants, while those with an AHI > 5 events/h were classified as OSA participants. Statistical analysis was performed using SPSS (v28, IBM Statistics, NY, USA) and GraphPad Prism (v10, GraphPad Software Inc., La Jolla, CA, USA). Normality was assessed by visual inspection of the Q–Q plots, and non-parametric test were used, or data were $\log_{10}$ transformed when necessary. To control for the impact of airway pressure on genioglossus multiunit activity, nadir $P_{epi}$ was included as a covariate in analyses. For clarity, the specific statistical

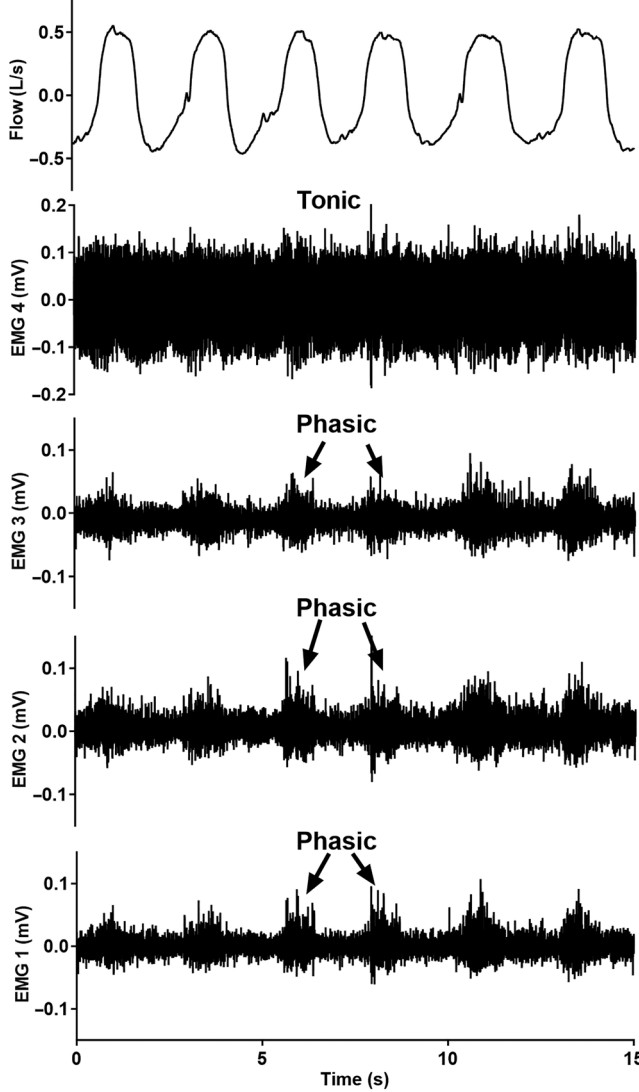

**Figure 2. Example of a single participant's EMG signals**
There were three tongue compartments (EMG 1–3) with phasic activity and one (EMG 4) with tonic activity. Signals were visually classified as either having a 'phasic' (respiratory modulation) or 'tonic' (no respiratory modulation) pattern of activity.

tests used for each comparison are included in the Results section. *P* values ≤0.05 were considered significant, except where the threshold was adjusted for multiple comparisons. In the tables, asterisks indicate a significant effect at *$P < 0.05$, **$P < 0.01$ and ***$P < 0.001$.

**Participant characteristics and respiratory parameters.** Mann–Whitney tests were used to evaluate differences in demographic variables and respiratory parameters between non-OSA and OSA groups. Fisher's exact test was used to test whether there was a difference in the sex proportion between OSA status groups. Spearman's correlations were used to evaluate the relationships between AHI and demographic and respiratory parameters with Bonferroni correction for multiple comparisons.

**Phasic and tonic EMG activity patterns.** First, a binary logistic regression analysis was used to test whether individual genioglossus compartments were more likely to have a phasic *versus* tonic activity pattern and whether the odds changed with OSA severity (i.e. AHI) while adjusting for nadir $P_{epi}$. The model was assessed for multicollinearity (largest variance inflation

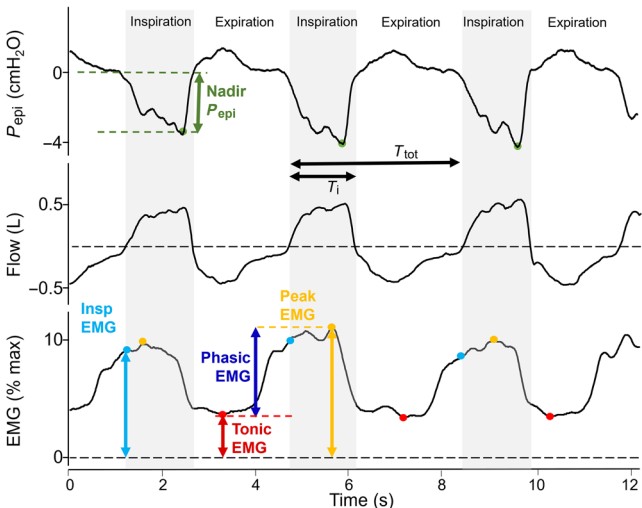

**Figure 3. Schematic showing the parameters derived from each breath**
Grey-shaded regions mark inspiration. $T_i$ (s) is the inspiration time and $T_{tot}$ (s) is the total breath time (inspiration + expiration). Nadir $P_{epi}$ (green dots) is the magnitude of the negative epiglottic pressure swing (i.e. decrease from baseline, defined as the point at which $P_{epi}$ becomes negative), and central inspiratory EMG (Insp EMG, blue dots) is the EMG amplitude at the beginning of inspiration. For each breath, peak EMG (yellow dots) is the maximum EMG amplitude, tonic EMG (red dots) is the minimum EMG amplitude, and phasic EMG (dark blue arrow) is the difference between peak EMG and nadir EMG. Rectified, moving time-averaged, EMG amplitudes are normalised to maximal EMG during tongue protrusion or swallow for analysis. [Colour figure can be viewed at wileyonlinelibrary.com]

factor = 2.045). Then, a mixed linear regression analysis was used to evaluate whether the maximum EMG used for the normalisation differed between tongue compartments. Finally, amongst subjects for whom four recordings were available, a mixed linear analysis was used to determine whether, for an individual, the number of wires with a phasic activity pattern (i.e. 0 – only tonic pattern, 1 to 3 – mixed tonic and phasic patterns, and 4 – only phasic pattern) was related to their OSA status and nadir $P_{epi}$.

**Quantitative EMG analysis for phasic genioglossus compartments.** In this section, only recordings with a phasic activity pattern were analysed since tonic EMG patterns are unlikely to be contributing to dynamic mechanisms to oppose the negative airway pressures occurring during inspiration. First, linear mixed models were used to assess the effect of genioglossus compartment on $log_{10}$-transformed peak, phasic, tonic and Insp EMG measurements while adjusting for nadir epiglottic pressure. Those were followed by Bonferroni-adjusted multiple comparisons to determine whether there were regional differences in the amplitude of the $log_{10}$-transformed peak, phasic, tonic and Insp EMG activity between compartments of the genioglossus (aim 1). Subsequently, another linear mixed model analysis was conducted to examine the interaction effect of genioglossus compartments × nadir $P_{epi}$ on $log_{10}$-transformed phasic EMG. Linear mixed model analyses included a random intercept for participants and the largest variance inflation factor was 1.871 for all.

**OSA severity effect on quantitative EMG measurements.** Across all phasically activated genioglossus compartments, the effect of OSA severity (AHI) on $log_{10}$-transformed peak, phasic, tonic and Insp EMG was assessed with a mixed linear model adjusted for nadir $P_{epi}$ (aim 2). The largest variance inflation factor was 1.221 for this model. Then, another mixed linear analysis was run to examine the interaction effects of genioglossus compartments × AHI on $log_{10}$-transformed phasic EMG, again while adjusting for nadir $P_{epi}$ (largest variance inflation factor = 1.848).

## Results

Data from nine participants were excluded due to poor quality or missing data (e.g. participants did not tolerate the insertion of any EMG wires or the epiglottic pressure catheter). As a result, data from 54 participants were analysed. Amongst those participants, incomplete EMG data were recorded from six participants, and their data were included in the analysis where available.

**Table 1. Participant characteristics and respiratory data**

| | Non-OSA | OSA | Mann–Whitney $U$ comparison |
|---|---|---|---|
| $N$ (M:F) | 9 (6:3) | 45 (34:11) | $P = 0.68$[#] |
| AHI (events/h) | $2.8 \pm 1.9$ [2.2, 0.5–5.0] | $27.5 \pm 20.3$ [23.8, 5.6–94.3] | $P < 0.001$*** |
| Age (years) | $46 \pm 15$ [46, 21–68] | $46 \pm 12$ [45, 20–73] | $P = 0.898$ |
| BMI (kg/m$^2$) | $25.0 \pm 3.4$ [24.2, 21.2–31.2] | $27.7 \pm 4.0$ [26.8, 21.5–40.4] | $P = 0.065$ |
| Nadir $P_{epi}$ (cmH$_2$O) | $-1.89 \pm 0.64$ [$-1.58$, $-2.93$ to $-1.19$] | $-3.71 \pm 1.74$ [$-3.51$, $-9.70$ to $-1.19$] | $P = 0.001$** |
| Inspiratory time, $T_i$ (s) | $1.77 \pm 0.30$ [1.80, 1.35–2.27] | $2.03 \pm 0.56$ [1.90, 1.31–3.85] | $P = 0.223$ |
| Total breath duration, $T_{tot}$ (s) | $4.22 \pm 0.91$ [4.10, 2.94–6.15] | $4.24 \pm 1.46$ [3.87, 2.69–11.83] | $P = 0.602$ |
| Breathing frequency, $F_b$ (bpm) | $15.0 \pm 2.9$ [14.9, 10.2–20.5] | $15.5 \pm 3.4$ [15.6, 5.3–22.4] | $P = 0.570$ |
| Tidal volume, $V_t$ (L) | $0.53 \pm 0.24$ [0.56, 0.11–0.90] | $0.89 \pm 0.49$ [0.72, 0.33–2.57] | $P = 0.021$* |
| Minute ventilation, $V_i$ (L/min) | $8.0 \pm 4.6$ [7.8, 1.5–18.5] | $12.8 \pm 5.9$ [11.1, 6.0–34.9] | $P = 0.011$* |
| Mean inspiratory flow rate, $V_t/T_i$ (L/s) | $0.30 \pm 0.16$ [0.28, 0.07–0.66] | $0.43 \pm 0.18$ [0.40, 0.20–1.00] | $P = 0.040$* |

Data are reported as mean $\pm$ SD [median, range]. Abbreviations: male (M), female (F), body mass index (BMI). [#]Fisher's exact test was used to assess the relationship between sex proportion and OSA status.

**Table 2. EMG activity patterns**

| Inspiratory EMG activity patterns (% maximum) | Phasic activity pattern $N = 106$ | Tonic activity patterns $N = 78$ |
|---|---|---|
| Peak EMG | $7.28 \pm 5.39$ [0.95–33.03] | – |
| Phasic EMG | $4.53 \pm 3.79$ [0.45–19.51] | – |
| Tonic EMG | $2.81 \pm 2.03$ [0.50–16.13] | $2.63 \pm 2.90$ [0.55–14.03] |
| Insp EMG | $4.24 \pm 3.24$ [0.11–22.86] | – |

Data are reported as mean $\pm$ standard deviation [minimum–maximum].

## Participant characteristics and respiratory parameters

A summary of participant characteristics and respiratory parameters for people with and without OSA is presented in Table 1. Sixteen participants had mild OSA (5 < AHI $\leq$ 15 events/h), 14 moderate OSA (15 < AHI $\leq$ 30 events/h) and 15 severe OSA (AHI > 30 events/h). Briefly, overall, the participants were typically middle-aged, and there were more males than females in both OSA and non-OSA groups. Those with OSA had more negative nadir $P_{epi}$, greater tidal volume, minute ventilation and greater mean inspiratory flow rate during supine and quiet nasal breathing than those without OSA. Other variables were not different between groups.

After accounting for the Bonferroni correction ($\alpha$ adjusted, $P < 0.0011$), a higher AHI was associated with a more negative nadir $P_{epi}$ (Spearman's rho $= -0.476$, $P < 0.001$) and a larger tidal volume (Spearman's rho $= 0.453$, $P < 0.001$).

## Phasic and tonic EMG activity patterns

EMG recordings were obtained for 184 wires in the 54 participants (85%) and averaged over $43 \pm 25$

[11–115] breaths in the anterior oblique ($n = 53$, 28.8%), posterior oblique ($n = 30$, 16.3%), anterior horizontal ($n = 41$, 22.3%) and posterior horizontal ($n = 60$, 32.6%) compartments of the tongue. The vast majority of wires were in OSA participants ($n = 157$, 85.3%).

Across the whole dataset, more than half of the EMG recordings were classified as phasic ($n = 106$, 57.6%). Tongue compartments with phasic EMG activity had significantly higher peak, phasic, tonic and Insp EMG than regions with tonic EMG (Table 2). Maximum EMG did not differ between compartments [mixed linear regression, $F(3,180) = 1.488$, $P = 0.219$].

Both tonic and phasic patterns of EMG were observed in all four tongue neuromuscular compartments. Figure 4 shows the distribution of patterns by compartments for both OSA and non-OSA participants. When adjusting for nadir $P_{epi}$, binary logistic regression showed no effect of OSA severity (AHI), tongue compartments and their interactions on the odds of the inspiratory EMG activity patterns of being phasic or tonic (Table 3).

Amongst participants for whom good quality recordings from all four wires were obtained ($n = 36$), a mixed linear analysis showed no effect of OSA status ($P = 0.123$) and nadir $P_{epi}$ ($P = 0.543$) on whether

**Table 3. Binary logistic regression showed no significant effect of OSA severity (AHI), genioglossus compartments or their interactions on the odds of the inspiratory EMG activity patterns being phasic or tonic when accounting for nadir $P_{epi}$**

| Inspiratory EMG activity patterns | | Exp (B) | 95% CI for exp (B) | *P* value |
|---|---|---|---|---|
| AHI (events/h) | | 0.994 | [0.960–1.029] | 0.742 |
| Nadir $P_{epi}$ (cmH$_2$O) | | 0.890 | [0.730–1.086] | 0.252 |
| Compartments: | | | | 0.469 |
| | Horizontal anterior | 0.895 | [0.233–3.436] | 0.872 |
| | Horizontal posterior | 0.503 | [0.128–1.973] | 0.325 |
| | Oblique anterior | 0.370 | [0.078–1.758] | 0.211 |
| AHI × compartments | | | | 0.937 |
| | AHI × Horizontal anterior | 0.999 | [0.958–1.041] | 0.948 |
| | AHI × Horizontal posterior | 0.998 | [0.956–1.043] | 0.931 |
| | AHI × Oblique anterior | 1.012 | [0.963–1.064] | 0.639 |
| Constant | | 0.817 | | 0.733 |

individuals had four genioglossus compartments with tonic patterns ($n = 4$, 11.1%), a mix of tonic and phasic patterns (i.e. $n = 8$, 22.2% had one compartment with phasic EMG, another $n = 8$, 22.2% had two compartments with phasic activity, and $n = 6$, 16.7% had three compartments with phasic activity), and four genioglossus compartments with phasic patterns ($n = 10$, 27.8%).

## Quantitative EMG analysis for phasic genioglossus compartments

Over all the phasically activated compartments, when adjusting for nadir $P_{epi}$ and accounting for repeated measures in participants, there was a significant effect of the genioglossus compartments on log$_{10}$-transformed peak, phasic and tonic, but not Insp EMG (Table 4). Pairwise comparisons showed that log$_{10}$-transformed

peak, phasic and tonic EMG were higher in the oblique than in the posterior horizontal compartments of the genioglossus (Fig. 5).

There was also a significant effect of nadir $P_{epi}$ for log$_{10}$-transformed phasic EMG only (Table 4), such that a more negative nadir $P_{epi}$ was associated with a higher EMG recording (Fig. 6A). However, in a secondary analysis examining the interaction effects of the genioglossus compartment × nadir $P_{epi}$ on the log$_{10}$-transformed phasic EMG, we found that only the anterior and posterior oblique compartments had a significant effect, but not the horizontal compartments (Table 5; Fig. 6B and C).

**OSA severity effect on quantitative EMG measurements.** Although there was no significant effect of AHI on the log$_{10}$-transformed peak, phasic, tonic and Insp EMG across all phasically active genioglossus

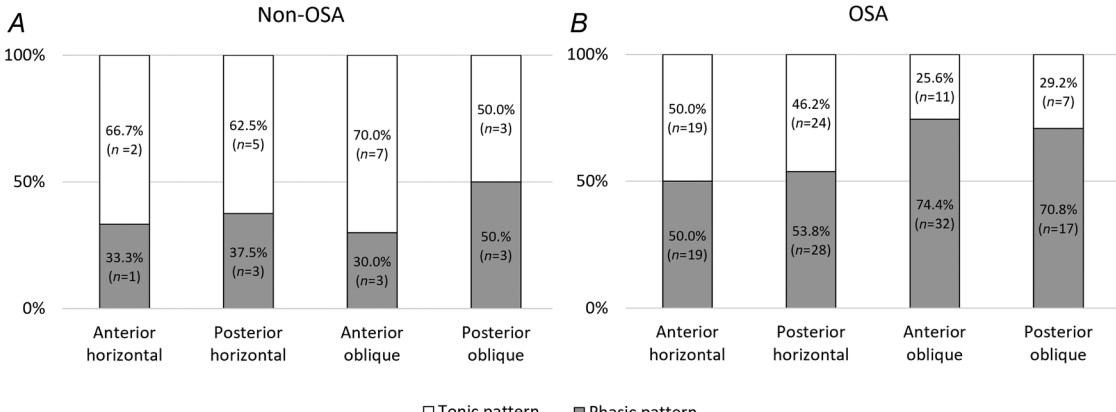

**Figure 4. Regional inspiratory EMG activity patterns**
The proportion of phasic (shaded) and tonic (unshaded) EMG activity patterns observed in each genioglossus neuromuscular compartment for (A) non-OSA participants (controls) and (B) OSA participants. When adjusting for nadir $P_{epi}$, there was no significant effect of OSA severity (AHI), tongue compartments and their interactions on the odds of the inspiratory EMG activity patterns being phasic or tonic (Table 3).

**Table 4. Estimates of fixed effects of the mixed linear regression to determine the effect tongue compartments on log$_{10}$-transformed peak, phasic, tonic and Insp EMG (dependent variables) for wires with phasic activity pattern only**

| Parameter | Log$_{10}$[Peak EMG (% of maximum)] |
|---|---|
| Intercept | Beta = 0.753, 95% CI [0.583 to 0.924], $t(101)$ = 8.761, $P < 0.001$ |
| Genioglossus compartments | |
| Horizontal anterior | Beta = −0.139, 95% CI [−0.332 to 0.054], $t(101)$ = −1.430, $P$ = 0.156 |
| Horizontal posterior | Beta = −0.267, 95% CI [−0.443 to −0.092], $t(101)$ = −3.018, $P$ = 0.003** |
| Oblique anterior | Beta = −0.009, 95% CI [−0.180 to 0.163], $t(101)$ = −0.101, $P$ = 0.920 |
| Oblique posterior | – |
| Nadir $P_{epi}$ (cmH$_2$O) | Beta = −0.031, 95% CI [−0.067 to 0.006], $t(101)$ = −1.672, $P$ = 0.098 |

| Parameter | Log$_{10}$[Phasic EMG (% of maximum)] |
|---|---|
| Intercept | Beta = 0.452, 95% CI [0.256 to 0.648], $t(101)$ = 4.565, $P < 0.001$ |
| Genioglossus compartments | |
| Horizontal anterior | Beta = −0.094, 95% CI [−0.316 to 0.129], $t(101)$ = −0.835, $P$ = 0.406 |
| Horizontal posterior | Beta = −0.277, 95% CI [−0.480 to −0.075], $t(101)$ = −2.714, $P$ = 0.008** |
| Oblique anterior | Beta = 0.033, 95% CI [−0.164 to 0.231], $t(101)$ = 0.335, $P$ = 0.738 |
| Oblique posterior | – |
| Nadir $P_{epi}$ (cmH$_2$O) | Beta = −0.043, 95% CI [−0.085 to −0.001], $t(101)$ = −2.014, $P$ = 0.047* |

| Parameter | Log$_{10}$[Tonic EMG (% of maximum)] |
|---|---|
| Intercept | Beta = 0.414, 95% CI [0.263 to 0.565], $t(101)$ = 5.425, $P < 0.001$ |
| Genioglossus compartments | |
| Horizontal anterior | Beta = −0.214, 95% CI [−0.385 to −0.043], $t(101)$ = −2.478, $P$ = 0.015* |
| Horizontal posterior | Beta = −0.226, 95% CI [−0.382 to −0.070], $t(101)$ = −2.876, $P$ = 0.005** |
| Oblique anterior | Beta = −0.071, 95% CI [−0.223 to 0.082], $t(101)$ = −0.920, $P$ = 0.360 |
| Oblique posterior | – |
| Nadir $P_{epi}$ (cmH$_2$O) | Beta = −0.022, 95% CI [−0.055 to 0.010], $t(101)$ = −1.356, $P$ = 0.178 |

| Parameter | Log$_{10}$[Insp EMG (% of maximum)] |
|---|---|
| Intercept | Beta = 0.569, 95% CI [0.374 to 0.764], $t(101)$ = 5.787, $P < 0.001$ |
| Genioglossus compartments | |
| Horizontal anterior | Beta = −0.037, 95% CI [−0.258 to 0.184], $t(101)$ = −0.332, $P$ = 0.741 |
| Horizontal posterior | Beta = −0.140, 95% CI [−0.341 to 0.061], $t(101)$ = −1.379, $P$ = 0.171 |
| Oblique anterior | Beta = 0.112, 95% CI [−0.084 to 0.308], $t(101)$ = 1.132, $P$ = 0.260 |
| Oblique posterior | – |
| Nadir $P_{epi}$ (cmH$_2$O) | Beta = 0.016, 95% CI [−0.026 to 0.058], $t(101)$ = 0.754, $P$ = 0.453 |

The model accounted for repeated measures in participants (random effect) and the covariate nadir epiglottic pressure (main effect). EMG measurements were log$_{10}$-transformed to improve data distribution normality.

**Table 5. Estimates of fixed effects of the mixed linear regression to determine the effect of the interactions tongue compartments × nadir $P_{epi}$ on log$_{10}$-transformed phasic EMG (dependent variables) for wires with phasic activity pattern only; the model accounted for repeated measures in participants (random effect)**

| Parameter | log$_{10}$ (Phasic EMG (% of maximum)) |
|---|---|
| Intercept | Beta = 0.357, 95% CI [0.198 to 0.516], $t(101)$ = 4.464, $P < 0.001$ |
| Horizontal anterior × nadir $P_{epi}$ | Beta = −0.036, 95% CI [−0.095 to 0.022], $t(101)$ = −1.242, $P$ = 0.217 |
| Horizontal posterior × nadir $P_{epi}$ | Beta = 0.001, 95% CI [−0.047 to 0.048], $t(101)$ = 0.025 $P$ = 0.980 |
| Oblique anterior × nadir $P_{epi}$ | Beta = −0.075, 95% CI [−0.122 to −0.027], $t(101)$ = −3.117, $P$ = 0.002** |
| Oblique posterior × nadir $P_{epi}$ | Beta = −0.080, 95% CI [−0.150 to −0.010], $t(101)$ = −2.252, $P$ = 0.027* |

Phasic EMG measurements were log$_{10}$-transformed to improve data distribution normality.

compartments (Table 6), a significant small effect of the OSA severity × compartment interaction was observed in the horizontal posterior compartment for the $\log_{10}$-transformed phasic EMG when adjusting for nadir $P_{epi}$ (Table 7 and Fig. 7). This effect was not observed in the other genioglossus compartments.

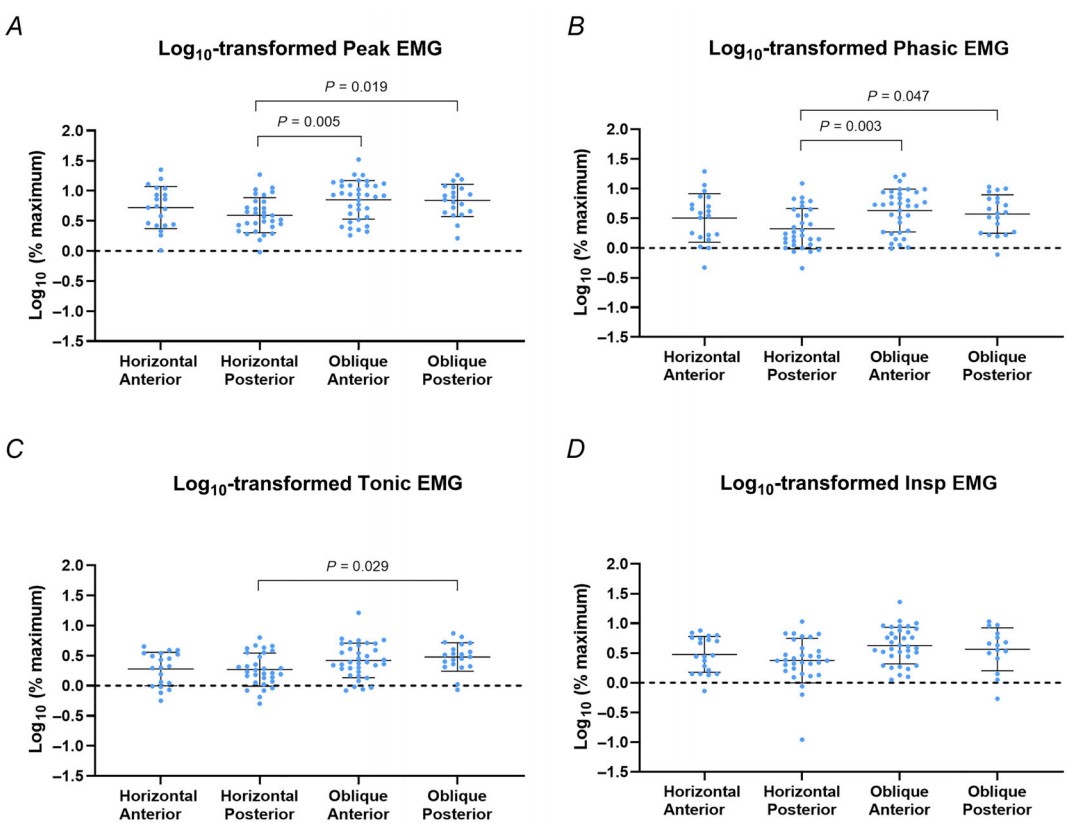

**Figure 5. $\log_{10}$-transformed EMG measurements for genioglossus compartments with phasic activity pattern**
Averaged across 43 ± 26 [12–115] breaths: *A*, peak EMG; *B*, phasic EMG; *C*, tonic EMG; and *D*, EMG at the onset of inspiration (Insp EMG). Means and standard deviations are superimposed. Significant pairwise comparisons (adjusted for Bonferroni multiple comparisons) are reported on the graphs. [Colour figure can be viewed at wileyonlinelibrary.com]

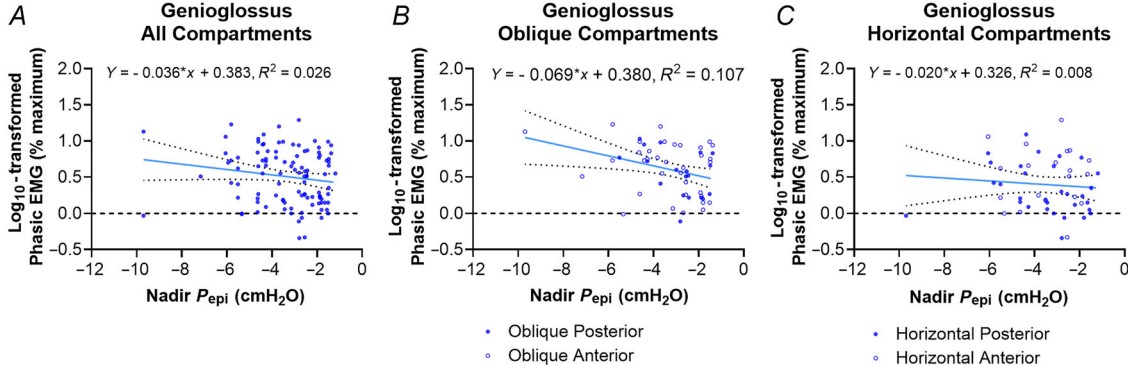

**Figure 6. Significant interactions between nadir $P_{epi}$ and the average $\log_{10}$-transformed phasic EMG averaged for each phasically active (*A*) genioglossus compartment, (*B*) oblique anterior and posterior compartments, and (*C*) horizontal anterior and posterior compartments (Tables 4 and 5)**
The linear regressions with 95% confidence intervals are plotted for each relationship, and the equations are reported along with the $R^2$ values. [Colour figure can be viewed at wileyonlinelibrary.com]

**Table 6. Estimates of fixed effects of the mixed linear regression to determine the effect of OSA severity (AHI) on log$_{10}$-transformed EMG measurements (dependent variables) across all genioglossus compartments with phasic activity pattern when adjusted for nadir $P_{epi}$; the model accounted for repeated measures in participants (random effect)**

| Parameter | Log$_{10}$[Peak EMG (% of maximum)] |
|---|---|
| Intercept | Beta = 0.682, 95% CI [0.538 to 0.825], $t(103)$ = 9.417, $P < 0.001$ |
| AHI (events/h) | Beta = −0.002, 95% CI [−0.006 to 0.002], $t(103)$ = −1.103, $P = 0.273$ |
| Nadir $P_{epi}$ (cmH$_2$O) | Beta = −0.034, 95% CI [−0.076 to 0.008], $t(103)$ = −1.618, $P = 0.109$ |

| Parameter | Log$_{10}$[Phasic EMG (% of maximum)] |
|---|---|
| Intercept | Beta = 0.401, 95% CI [0.236 to 0.566], $t(103)$ = 4.820, $P < 0.001$ |
| AHI (events/h) | Beta = −0.002, 95% CI [−0.006 to 0.002], $t(103)$ = −1.128, $P = 0.262$ |
| Nadir $P_{epi}$ (cmH$_2$O) | Beta = −0.048, 95% CI [−0.097 to 0.000], $t(103)$ = −1.989, $P = 0.049$* |

| Parameter | Log$_{10}$[Tonic EMG (% of maximum)] |
|---|---|
| Intercept | Beta = 0.319, 95% CI [0.193 to 0.445], $t(103)$ = 5.032, $P < 0.001$ |
| AHI (events/h) | Beta = −0.002, 95% CI [−0.005 to 0.001], $t(103)$ = -1.145, $P = 0.255$ |
| Nadir $P_{epi}$ (cmH$_2$O) | Beta = −0.025, 95% CI [−0.061 to 0.012], $t(103)$ = −1.322, $P = 0.189$ |

| Parameter | Log$_{10}$[Insp EMG (% of maximum)] |
|---|---|
| Intercept | Beta = 0.581, 95% CI [0.421 to 0.741], $t(101)$ = 7.219, $P < 0.001$ |
| AHI (events/h) | Beta = −0.002, 95% CI [−0.006 to 0.002], $t(101)$ = −1.148, $P = 0.254$ |
| Nadir $P_{epi}$ (cmH$_2$O) | Beta = 0.006, 95% CI [−0.040 to 0.053], $t(101)$ = 0.273, $P = 0.785$ |

Phasic EMG measurements were log$_{10}$-transformed to improve data distribution normality.

**Table 7. Estimates of fixed effects of the mixed linear regression to determine the effect of the genioglossus compartment and OSA severity (AHI) interactions on log$_{10}$-transformed phasic EMG measurements (dependent variables) across all genioglossus compartments with phasic activity pattern when adjusted for nadir $P_{epi}$; the model accounted for repeated measures in participants (random effect)**

| Parameter | Log$_{10}$[Phasic EMG (% of maximum)] |
|---|---|
| Intercept | Beta = 0.381, 95% CI [0.220 to 0.543], $t(100)$ = 4.676, $P < 0.001$ |
| Horizontal anterior × AHI | Beta = −0.001, 95% CI [−0.007 to 0.006], $t(100)$ = −0.173, $P = 0.863$ |
| Horizontal posterior × AHI | Beta = −0.007, 95% CI [−0.012 to −0.002], $t(100)$ = −2.820, $P = 0.006$** |
| Oblique anterior × AHI | Beta = 0.001, 95% CI [−0.004 to 0.007], $t(100)$ = 0.559, $P = 0.577$ |
| Oblique posterior × AHI | Beta = 0.000, 95% CI [−0.007 to 0.007], $t(100)$ = −0.032, $P = 0.974$ |
| Nadir $P_{epi}$ (cmH$_2$O) | Beta = −0.051, 95% CI [−0.098 to −0.004], $t(100)$ = −2.141, $P = 0.035$* |

Phasic EMG measurements were log$_{10}$-transformed to improve data distribution normality.

## Discussion

### Main findings

This study investigated whether there were compartmental differences in multiunit EMG activity in the anterior and posterior sections of the horizontal and oblique compartments of the genioglossus during quiet nasal breathing in supine awake non-obese individuals with and without OSA. There are four main findings. First, both tonic and phasic activity patterns were observed in those with and without OSA during quiet breathing, and when a compartment was phasically active (∼58% of compartments), phasic EMG varied from ∼1% to 20% of maximum, suggesting that the genioglossus is differentially activated between individuals. Second, in genioglossus compartments with a phasic activity pattern, peak, phasic and tonic EMG quantitative measurements were higher in the oblique than the horizontal genioglossus, indicating compartmental variation in genioglossus activity during awake inspiration. Third, greater phasic activity was associated with more negative nadir $P_{epi}$ in the oblique compartments, consistent with reflex activation of the genioglossus (Carberry et al., 2022; Horner et al., 1991). Fourth, there were no differences in genioglossus activation with OSA severity, except for a small effect in

the horizontal posterior compartment, suggesting that people with and without OSA exhibit a similar range of drive during wakefulness. These results do not support our hypotheses.

## Phasic and tonic EMG genioglossus activation and respiratory effort

Genioglossus EMG activity during inspiration is driven by both phasic and tonic inputs from the brainstem respiratory control centres, summed with reflex excitation mediated by airway pressure mechanoreceptors (Butler, 2007; Carberry et al., 2015). Typically, the neural drive to breathe increases during inspiration and decreases during expiration, resulting in a phasic activity pattern. However, our results show that phasic activity was absent in ∼42% of the genioglossus compartments. It may be that compartments lacking phasic activity during quiet breathing may not be actively contributing to airway dilatation during quiet breathing. Instead, tonic EMG could act to stiffen the airway wall instead. Indeed, the relationship between genioglossus EMG and airway dilatation in OSA is not direct. Our recent work showed that tongue dilatory movement can occur in the presence of minimal genioglossus phasic activity (Juge et al., 2023) or that genioglossus EMG may not result in dilatation.

Therefore, we speculate that those compartments with only tonic EMG may not have reached the threshold for recruitment of phasic activity during quiet breathing in wakefulness. Supporting this are two lines of evidence. First, it is well-established that the genioglossus has inspiratory phasic motor units whose activity increases during inspiration (Saboisky et al., 2007), and there

appears to be some variation in the timing of the inspiratory activation across the respiratory cycle, suggesting a dynamic drive from both central inspiratory and reflexive upper airway negative pressure feedback. Second, two previous studies have reported a mix of multiunit EMG tonic and phasic genioglossus activity during quiet awake breathing (Eastwood et al., 2003; Vranish & Bailey, 2015). Those studies also showed that the tonic genioglossus region became phasic as breathing effort increased, suggesting that phasic activity across genioglossus regions increases with increased drive to breathe and increased pharyngeal pressure, although no people with OSA were included.

We also found that OSA severity did not have an additive effect on phasic EMG on top of a more negative $P_{epi}$, suggesting the neural control mechanisms governing genioglossus activation remain largely intact in people with OSA. This is consistent with previous work, which did not find any major difference in the pattern of firing rates of genioglossus motor unit activity (Luu et al., 2020; Saboisky et al., 2007) or genioglossus multiunit peak and tonic EMG (Oliven et al., 2018) in people with or without OSA. It seems likely that the anatomical factors that predispose people to OSA, such as obesity and narrow airways (Isono, 2012), cause greater airway resistance and result in a more negative nadir $P_{epi}$, which in turn results in greater compensatory phasic EMG during supine wakefulness. This is supported by the observation in our sample that nadir $P_{epi}$ was typically less negative in people without OSA (i.e. ∼−2 cmH$_2$O) than in participants with OSA (∼−3.7 cmH$_2$O, Table 1) and could explain in part why previous studies reported greater genioglossus multiunit EMG in obese OSA participants with an AHI > 25 events/h than in controls during wakefulness (Fogel et al., 2001; Mezzanotte et al., 1992). Indeed, the mean body mass index (BMI) of the participants in the current study was in the non-obese range, whereas the BMI of the participants in the Mezzanotte et al. study was in the morbidly obese range. Thus, the current findings may not extend to more obese participants with greater negative epiglottic pressure swings during quiet breathing.

Finally, we observed a wide range in the magnitude of genioglossus phasic EMG due to our inclusion of people with OSA, some of whom were obese. For comparison, in healthy adults, Vranish & Bailey (2015) reported inspiratory multiunit EMG of ∼6% during quiet breathing, ∼8% during deep breathing and ∼15% during a Mueller manoeuvre and hyperventilation, where breathing effort was intentionally increased. This highlights the wider spectrum of baseline respiratory genioglossus activity in those individuals with OSA, which was not observed in studies limited to healthy populations.

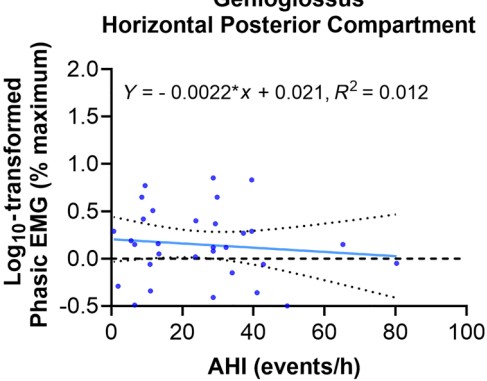

**Figure 7. Significant interactions between OSA severity (AHI) and the log$_{10}$-transformed phasic EMG averaged for each phasically active horizontal posterior genioglossus compartment (Table 7)**

The linear regressions with 95% confidence intervals are plotted for each relationship, and the equations are reported along with the $R^2$ values. [Colour figure can be viewed at wileyonlinelibrary.com]

## Functional compartmentalisation of genioglossus drive

Our results support functional compartmentalisation of genioglossus drive during quiet breathing in wakefulness, as oblique compartments exhibited a higher activity than horizontal compartments. The oblique compartment is typically positioned behind the soft palate, which is often the narrowest part of the airway. Functionally, this increased activation may contribute to maintaining airway patency by providing greater support in regions more susceptible to collapse. This aligns with previous research suggesting that different regions of the tongue contribute differently to airway patency (Bilston & Gandevia, 2014).

The tongue is a muscular hydrostat (Kier & Smith, 1985), capable of precise movements via shape adjustments to meet the various functional demands associated with respiration, as well as speech and swallowing. These adjustments occur via different contraction patterns of the tongue muscles, including the four functional compartments of the genioglossus. Supporting this concept, task-specific regional activation of the genioglossus has been observed during protrusion and swallowing (Yeung et al., 2022). It is thought to be enabled by the different hypoglossal nerve branches innervating the horizontal and oblique neuromuscular compartments of the genioglossus (Mu & Sanders, 2010). Furthermore, the genioglossus muscle is known for its anatomical heterogeneity. The anterior area often contains more type II muscle fibres (Saigusa et al., 2001), the posterior area has more inspiratory motor units (Luu et al., 2020) and the horizontal posterior section tends to have more fat deposition (Jugé et al., 2021; Kim et al., 2014). Muscle fibre arrangements also differ between the horizontal and oblique compartments (Sanders & Mu, 2013). Imaging studies have shown heterogeneous dilatory patterns across the genioglossus during breathing (Brown et al., 2013; Jugé et al., 2020), indicating that there is a dissociation between dilatory movement of the tongue and neural drive in a substantial proportion of people with OSA (Juge et al., 2023). These studies, together with our current results, which found no differences in EMG activation patterns but a difference in amplitudes between genioglossus compartments, support the concept that the biomechanical output of the tongue during breathing cannot be solely predicted by EMG output and that anatomy and local biomechanics must also be taken into account.

## EMG genioglossus activation at the beginning of inspiration and implications for OSA pathophysiology

'Insp EMG' was measured at the onset of inspiration when airflow started to increase. During wakefulness, activity recorded in the genioglossus begins ∼100 ms before the onset of inspiration (Butler, 2007; Strohl et al., 1980), reflecting central inspiratory drive from the respiratory pattern generator prior to the negative pressure swings in the pharynx that recruit additional reflexive dilatory EMG activity. The contribution of central inspiratory drive can therefore be partly examined during the temporal window prior to these pharyngeal negative pressure swings that occur during mid- to late inspiration. Differentiating between central inspiratory drive and negative pressure reflex mechanisms is crucial for understanding their respective contributions (Pillar et al., 2001). Here, the magnitude of Insp EMG in genioglossus compartments with phasic activity pattern shows that neural drive to the genioglossus was present at the onset of airflow, indicating a substantial contribution of the central drive to phasic genioglossus activation in those compartments. This is consistent with observations of anterior tongue movement (Cheng et al., 2011) and recruitment of genioglossus motor units in people with and without OSA (Saboisky et al., 2007) preceding inspiratory flow.

The presence of central respiratory drive prior to the onset of inspiration in phasically activated genioglossus compartments suggests that the central respiratory pattern generator plays a pivotal role in preparing the airway for the negative pressure associated with airflow during inhalation during wakefulness (Messineo et al., 2022), although the role of this preparatory drive in OSA pathophysiology is not well understood. It may also increase sensitivity to reflex drive as those compartments also respond to negative airway pressure by reflex activation. If confirmed, this might have implications for individuals with OSA. A reduced central drive at sleep onset could predispose the upper airway to collapse due to insufficient 'preparatory' central activation, especially if reflex compensation mechanisms are not sufficient to maintain the airway opening (McGinley et al., 2008). Understanding the role of central drive in maintaining airway patency could inform novel interventions aimed at enhancing preparatory muscle activation, such as hypoglossal nerve stimulation or targeted myofunctional therapy, to improve airway stability during sleep.

## Limitations

The first limitation of this study was that once the EMG recordings were grouped into OSA and non-OSA categories and the tonic pattern was removed from the quantitative analysis, there were small numbers in some groups, limiting statistical power for analysis of the quantitative EMG measures by OSA status. We also had only nine control participants (*vs.* 45 with OSA). Although these are still relatively large numbers for invasive physiology studies (Horner et al., 1991; Mezzanotte et al., 1992), greater numbers would allow further categorisation of

the compartmental heterogeneity of upper airway control (Brown et al., 2013; Carberry et al., 2022; Eastwood et al., 2003; Juge et al., 2020; Vranish & Bailey, 2015; Yeung et al., 2022). Second, awake data were collected, as measuring genioglossus activity during the awake state provides valuable insights into the baseline neuromuscular properties and compensatory mechanisms that may differ between individuals with and without OSA, including those that are not sleep-dependent. However, it is known that the sleep state modifies central output to the genioglossus and attenuates upper airway reflexes (Cori et al., 2018; Gell et al., 2022; Sauerland & Harper, 1976). Therefore, further work is needed to investigate compartmental drive during different sleep stages. Third, accurately locating the electrode in specific genioglossus neuromuscular compartments is a challenge, as the border of the oblique and horizontal compartments cannot be visualised by ultrasound or detected during wire insertion. Here, the placement was confirmed by physiological (functional) responses during swallowing (Yeung et al., 2022). Fourth, we cannot exclude the possibility that the differential levels of genioglossus activation observed between individuals during inspiration may, at least in part, be influenced by the positioning of the recording electrodes relative to the local firing motor units. Given that EMG recordings capture activity from a localised muscle volume surrounding the electrode tip, variations in electrode placement could contribute to the observed differences. Fifth, positioning the posterior electrodes further back could have revealed larger differences between the anterior and posterior compartments. However, our setup followed similar multielectrode recording approaches used by us (Carberry et al., 2022; Yeung et al., 2022) and others (Eastwood et al., 2003). This placement ensured reliable data while minimising risks, such as less precise ultrasound-guided placement in more posterior regions (especially in larger tongues, for which the back of the tongue might not be seen) and reduced electrode stability due to increased oropharyngeal movement. Future studies could explore alternative placements to further investigate this hypothesis. Lastly, participants with OSA had a mean BMI of almost 3 kg/m$^2$ higher than the controls but still within the non-obese range. Although no statistically significant difference between groups was observed, this may have influenced comparisons between groups, as the OSA group breathed with a larger tidal volume than the control group, and ventilation patterns may influence genioglossus activity. By including nadir $P_{epi}$ as a covariate in our models, we aimed to control for this effect and reduce its potential impact on group comparisons.

## Conclusions

Muscle activity of the anatomically distinct horizontal and oblique compartments of the genioglossus during awake and supine quiet breathing can be either phasic ($\sim$60%) or tonic ($\sim$40%) and varies markedly between individuals. We initially hypothesised that muscle activation of the horizontal compartment would be greater than that in the oblique compartment, but we found the opposite. This suggests that during wakefulness, neural drive to the genioglossus does not produce a uniform or coordinated activation pattern between compartments but rather relies on a varied regional activation in response to changing respiratory demands to maintain airway patency. We also hypothesised that EMG genioglossus activation would be greater in OSA participants than in non-OSA participants. However, this was not the case in this cohort, who were, on average, not obese. Instead, our results show nadir $P_{epi}$ was associated with phasic genioglossus activity, indicating that diaphragm activation required to maintain airway patency is a key factor in the magnitude of the genioglossus activity in people with and without OSA.

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

## Additional information

### Data availability statement

Data are available only on request due to privacy/ethical restrictions.

### Competing interests

Outside the submitted work, D.J.E. has had research grants from Bayer, Takeda, Invicta Medical and Apnimed and has served on Scientific Advisory Boards for Apnimed, Invicta and Mosanna and as a consultant for Bayer. None of the other authors has any conflicts of interests.

### Author contributions

Lauriane Jugé: Acquisition, analysis or interpretation of data for the work; Drafting the work or revising it critically for important intellectual content; Final approval of the version to be published; Agreement to be accountable for all aspects of the work Peter Burke: Acquisition, analysis or interpretation of data for the work; Drafting the work or revising it critically for important intellectual content; Final approval of the version to be published; Agreement to be accountable for all aspects of the work Jade Yeung: Acquisition, analysis or interpretation of data for the work; Drafting the work or revising it critically for important intellectual content; Final approval of the version to be published; Agreement to be accountable for all aspects of the work Fiona Knapman: Acquisition, analysis or interpretation of data for the work; Drafting the work or revising it critically for important intellectual content; Final approval of the version to be published; Agreement to be accountable for all aspects of the work Elizabeth Brown: Acquisition, analysis or interpretation of data for the work; Drafting the work or revising it critically for important intellectual content; Final approval of the version to be published; Agreement to be accountable for all aspects of the work Alan Chiang: Acquistion, analysis or interpretation of data for the work; Drafting the work or revising it critically for important intellectual content; Final approval of the version to be published; Agreement to be accountable for all aspects of the work Danny Eckert: Acquisition, analysis or interpretation of data for the work; Drafting the work or revising it critically for important intellectual content; Final approval of the version to be published; Agreement to be accountable for all aspects of the work Jane Butler: Conception or design of the work; Acquisition, analysis or interpretation of data for the work; Drafting the work or revising it critically for important intellectual content; Final approval of the version to be published; Agreement to be accountable for all aspects of the work Lynne Bilston: Conception or design of the work; Acquisition, analysis or interpretation of data for the work; Drafting the work or revising it critically for important intellectual content; Final approval of the version to be published; Agreement to be accountable for all aspects of the work

## Funding

This research was funded by the National Health & Medical Research Council (NHMRC) of Australia (#APP1058974). L.E.B., D.J.E. and J.E.B. are supported by NHMRC Fellowships (#APP1077934, APP1196261 and APP1042646, respectively).

## Acknowledgements

The authors acknowledge Dr Peter Humburg, statistical consultant at the University of New South Wales Mark Wainwright Analytical Centre (Stats Central), for his valuable support with the statistical analysis of this study.

Open access publishing facilitated by University of New South Wales, as part of the Wiley - University of New South Wales agreement via the Council of Australian University Librarians.

## Keywords

intramuscular tongue EMG, respiratory physiology, sleep-disordered breathing, upper airway

## Supporting information

Additional supporting information can be found online in the Supporting Information section at the end of the HTML view of the article. Supporting information files available:

**Peer Review History**

