## [Peer Review History · The Journal of Physiology]

Compartmental inspiratory genioglossus electromyographic activity in supine, awake individuals with and without obstructive sleep apnoea

Lauriane Jugé, Peter GR Burke, Jade Yeung, Fiona L Knapman, Elizabeth C. Brown, Alan KI Chiang, Danny J Eckert, Jane E Butler, and Lynne E Bilston

DOI: 10.1113/JP287943

Corresponding author(s): Lynne Bilston (l.bilston@unsw.edu.au)

The following individual(s) involved in review of this submission have agreed to reveal their identity: Ron Oliven (Referee #1)

Review Timeline:	Submission Date:	23-Oct-2024
	Editorial Decision:	02-Jan-2025
	Revision Received:	25-Feb-2025
	Accepted:	25-Mar-2025

Senior Editor: Harold Schultz

Reviewing Editor: Ken O'Halloran

Transaction Report:

Dear Dr Bilston,

Re: JP-RP-2024-287943 "Compartmental inspiratory genioglossus electromyographic activity in supine, awake individuals with and without obstructive sleep apnoea" by Lauriane Jugé, Peter GR Burke, Jade Yeung, Fiona L Knapman, Elizabeth C. Brown, Alan KI Chiang, Danny J Eckert, Jane E Butler, and Lynne E Bilston

Thank you for submitting your manuscript to The Journal of Physiology. It has been assessed by a Reviewing Editor and by 2 expert referees and we are pleased to tell you that it is acceptable for publication following satisfactory revision.

REVISION CHECKLIST:

We look forward to receiving your revised submission.

Yours sincerely,

Harold Schultz
Senior Editor
The Journal of Physiology

REQUIRED ITEMS

- Author photo and profile. First or joint first authors are asked to provide a short biography (no more than 100 words for one author or 150 words in total for joint first authors) and a portrait photograph. These should be uploaded and clearly labelled together in a Word document with the revised version of the manuscript. See Information for Authors for further details.
- The Journal of Physiology funds authors of provisionally accepted papers to use the premium BioRender site to create high resolution schematic figures. Follow this link and enter your details and the manuscript number to create and download figures. Upload these as the figure files for your revised submission. If you choose not to take up this offer, we require figures to be of similar quality and resolution. If you are opting out of this service to authors, state this in the Comments section on the Detailed Information page of the submission form. The link provided should only be used for the purposes of this submission. Authors will be charged for figures created on this premium BioRender account if they are not related to this manuscript submission.
- Please upload separate high-quality figure files via the submission form.
- Please ensure that the Article File you upload is a Word file.
- Papers must comply with the Statistics Policy: https://jp.msubmit.net/cgi-bin/main.plex?form_type=display_requirements#statistics.

In summary:

- If n {less than or equal to} 30, all data points must be plotted in the figure in a way that reveals their range and distribution. A bar graph with data points overlaid, a box and whisker plot or a violin plot (preferably with data points included) are acceptable formats.
- If $n > 30$, then the entire raw dataset must be made available either as supporting information, or hosted on a not-for-profit repository, e.g. FigShare, with access details provided in the manuscript.
- 'n' clearly defined (e.g. x cells from y slices in z animals) in the Methods. Authors should be mindful of pseudoreplication.

- All relevant 'n' values must be clearly stated in the main text, figures and tables.
 - The most appropriate summary statistic (e.g. mean or median and standard deviation) must be used. Standard Error of the Mean (SEM) alone is not permitted.
 - Exact p values must be stated. Authors must not use 'greater than' or 'less than'. Exact p values must be stated to three significant figures even when 'no statistical significance' is claimed.
-
- Please include an Abstract Figure file, as well as the Figure Legend text within the main article file. The Abstract Figure is a piece of artwork designed to give readers an immediate understanding of the research and should summarise the main conclusions. If possible, the image should be easily 'readable' from left to right or top to bottom. It should show the physiological relevance of the manuscript so readers can assess the importance and content of its findings. Abstract Figures should not merely recapitulate other figures in the manuscript. Please try to keep the diagram as simple as possible and without superfluous information that may distract from the main conclusion(s). Abstract Figures must be provided by authors no later than the revised manuscript stage and should be uploaded as a separate file during online submission labelled as File Type 'Abstract Figure'. Please also ensure that you include the figure legend in the main article file. All Abstract Figures should be created using BioRender. Authors should use The Journal's premium BioRender account to export high-resolution images. Details on how to use and access the premium account are included as part of this email.
-

Reviewing Editor's comments:

Thank you for submitting your manuscript to The Journal of Physiology. Two experts in the field have reviewed your work and I am pleased to report that both are enthusiastic about the merits of the paper and its potential influence for the wider field. Both referees have suggestions that should be carefully considered. In general these encourage the authors to view the data through different lenses to ensure the most comprehensive considerations are included in the final manuscript. I look forward to receiving your responses to the referees and a revised manuscript.

Senior Editor's comments:

Thank you for submission of your research article to the Journal of Physiology for consideration. The article has been

reviewed by experts in the field and found to be potentially acceptable for publication pending adequate revision to address all of the concerns raised. Please address all comments from the external referees and reviewing editor as well as addressing the list of requirements or publication in the journal included in this letter and the link below.

<https://physoc.onlinelibrary.wiley.com/pb-assets/hub-assets/physoc/documents/TJP-Rigour-and-Reproducibility-Requirements-1724673661727.pdf>

Referee #1:

This study builds on several prior publications from the same research group, utilizing a methodology in which they have considerable expertise to map the distribution of neural drive to the genioglossus (GG) in individuals with and without obstructive sleep apnea (OSA). The authors found no significant differences in electromyographic (EMG) activity between the different compartments of the GG. Likewise, no differences were observed between OSA patients and healthy subjects.

The manuscript is well-written and makes a significant contribution to the existing body of knowledge on the neural control of the GG. The methodology is thoroughly described, and despite the complexity of the results, they are presented in a clear and comprehensible manner. The authors have also appropriately discussed the important limitations of the study. I have only minor comments and suggestions for consideration:

1. Electrodes placement: According to Figure 1, the posterior-segment electrodes were placed relatively close to the anterior ones, effectively in the middle of the GG. It is possible that placing these electrodes further posteriorly might reveal larger differences between the anterior and posterior compartments.
2. Table 1: Why were the OSA patients hyperpneic, with high tidal volumes? This ventilation pattern could influence EMG activity, which is often linked to diaphragm activity, and also affect Pepi. Could this observation impact the comparison between participants with and without OSA, and might it warrant statistical correction?
3. OSA severity range: The authors should provide the number of OSA patients in each severity range (mild, moderate, severe). If the sample includes a substantial proportion of patients with mild OSA, the authors should consider comparing normal controls to patients with moderate and severe OSA only (or apply an alternative statistical approach) to evaluate a potential relationship between EMG activity and OSA severity.
4. Figure 4: The lack of statistically significant differences might be partly due to the statistical analysis used, which integrated all data. For instance, when examining the regional inspiratory activity pattern of the anterior oblique compartment in OSA and non-OSA groups, the difference appears significant despite the small number of controls. Please comment on this observation.
5. Awake state recordings: The authors recognize that limiting EMG recordings to the awake state is a limitation, as the neural control of the GG changes significantly during sleep. However, they should also clarify the relevance of comparing awake GG activity during relaxed breathing in subjects with and without OSA.
6. Pre-inspiratory inspiratory activity: As noted by the authors, the pre-inspiratory inspiratory activity of upper airway dilators and its significance has been recognized for more than 40 years. Nevertheless, the role of this central "preparatory" drive in OSA pathophysiology has not been validated. Since this topic is beyond the scope of the current study, the discussion on this subject could be abbreviated. Conversely, the authors might consider suggesting potential implications of their findings for recent treatment modalities, such as hypoglossal stimulation and myofunctional therapy.

Referee #2:

Deficits in genioglossus reflex activity are key to the development of obstructive sleep apnea, but detailed physiology studies are lacking. In the current study, Jugé and colleagues studied a large number of OSA patients and healthy controls during quiet breathing to assess whether 1), there are differences in genioglossus EMG between oblique and horizontal compartments and 2), whether compartment-specific genioglossus EMG activation is different in OSA versus controls. As part of the rationale, the authors expected that EMG would be greater in the horizontal vs oblique compartments, and that EMG activation would be greater in OSA versus controls (compensation for the greater negative pharyngeal pressure in OSA). In analyses adjusting for the epiglottic pressure swings, the authors found that there were no differences in genioglossus EMG levels between oblique and horizontal compartments. The authors also found no adjusted differences between OSA and controls in genioglossus EMG levels.

I have several recommendations to improve the presentation and focus on the primary aims.

- 1) To address their Aim 1, the authors provided figures showing raw sample data with means/SD (Figure 5, then adjusted mixed model F-values and P-values (Table 4), but did not provide adjusted EMG (least-squares) differences (and 95%CI) across compartments. Providing beta values for each of the compartments (there should be three difference terms) is needed to interpret the similarities in genioglossus EMG; likewise 95%CI are needed to interpret the stability of the estimates.
- 2) Based on the figures, it seems like there may be a quantitatively meaningful increase in peak activity in oblique compared to horizontal (particularly horizontal). I encourage the authors to more carefully present the results relating to their primary hypothesis. The primary model presented in Table 4 may be over interpreted: The model describes a multi-level term for compartment, OSA status, and a compartment-by-OSA status interaction term. The inclusion of the interaction term, therefore means that the main effect for compartment, as presented, is based on data primarily from controls (which is considerably more limited in sample). I suggest first presenting results without inclusion of the OSA status term.
- 3) The rationale for removing phasic components from an analysis of the strength of phasic EMG activity is not entirely clear. For example, the horizontal compartments appear to be less-often phasic (Figure 4) and the phasic components also appear to have less activity; if the authors instead chose to combine tonic and phasic compartments (in what I would see as a less biased analysis) then they may have found smaller EMG activity on average in the horizontal vs oblique compartments. I suggest not parsing out compartments by tonic/phasic status, at least in the primary analyses.
- 4) In the assessment of OSA versus controls, I suggest performing analysis without and then with Pepi adjustment: It would be important to know if EMG in OSA is > EMG in controls for comparison to the prior Mezzinotte/Malhotra papers in the literature, then to see OSA-vs-control differences disappear with Pepi adjustment. The rationale also alludes to Pepi differences as being a reason for OSA related increase in GG activity so adjusting Pepi effects out in primary analysis appears unhelpful.
- 5) I suggest considering performing an exploratory analysis that uses separate terms for anterior/posterior status, and horizontal/oblique status (replacing the 4-level term). This analysis may have more statistical power due to fewer terms, and also appears more consistent with the introductory rationale. Note that the anterior/posterior status appears less important than horizontal/oblique status.
- 6) EMG results appear not-normally distributed, you may like to square-root transform results before analysis.
- 7) The authors used %max for analysis, but the max maneuvers may have activated some compartments more than others. Have the authors considered using data in uV units, or considered adjusting for the %max calibration uV levels.
- 8) To understand if "genioglossus responsiveness" depends on compartment type, the authors may like to see include a compartment-by-Pepi term to test whether compartments differ in their response to negative pressure (suggest not including OSA status in this analysis due to unnecessary complexity).

Minor:

-Beta values are missing units, are these %max/cmH₂O or uV/cmH₂O?

Table 4 illustrates a multivariable regression but shows only F and P values rather than betas and 95%CI, which makes it difficult to interpret if there are any physiologically-meaningful differences present that may have not reached significance. Units are missing (%max?). e.g. There are four GG compartments, yet we can't see the beta for each of them, only the F value for all of them combined.

Figure 6. too many decimal points in the values within the figures.

Re: ". These results reject our hypotheses."

-Technically your results did not support your hypotheses, please rephrase.

END OF COMMENTS

JP-RP-2024-287943

"Compartmental inspiratory genioglossus electromyographic activity in supine, awake individuals with and without obstructive sleep apnoea"

by Lauriane Jugé, Peter GR Burke, Jade Yeung, Fiona L Knapman, Elizabeth C. Brown, Alan KI Chiang, Danny J Eckert, Jane E Butler, and Lynne E Bilston

REQUIRED ITEMS

- Author photo and profile. First or joint first authors are asked to provide a short biography (no more than 100 words for one author or 150 words in total for joint first authors) and a portrait photograph. These should be uploaded and clearly labelled together in a Word document with the revised version of the manuscript. See Information for Authors for further details.

Dr Lauriane Jugé and Dr Peter Burke (co-first authors) have provided a short biography and portrait photos.

- The Journal of Physiology funds authors of provisionally accepted papers to use the premium BioRender site to create high resolution schematic figures. Follow this link and enter your details and the manuscript number to create and download figures. Upload these as the figure files for your revised submission. If you choose not to take up this offer, we require figures to be of similar quality and resolution. If you are opting out of this service to authors, state this in the Comments section on the Detailed Information page of the submission form. The link provided should only be used for the purposes of this submission. Authors will be charged for figures created on this premium BioRender account if they are not related to this manuscript submission.

BioRender was used to prepare a high-quality abstract figure.

- Please upload separate high-quality figure files via the submission form.

Seven High-resolution Figure versions have been uploaded and prepared with a width of either 17.5 or 8.5 cm. Figures captions are listed at the end of the article file.

- Please ensure that the Article File you upload is a Word file.

The manuscript uploaded is in Word format with seven tables embedded.

- Papers must comply with the Statistics Policy: https://jp.msubmit.net/cgi-bin/main.plex?form_type=display_requirements#statistics.

In summary:

- If n {less than or equal to} 30, all data points must be plotted in the figure in a way that reveals their range and distribution. A bar graph with data points overlaid, a box and whisker plot or a violin plot (preferably with data points included) are acceptable formats.

- If $n > 30$, then the entire raw dataset must be made available either as supporting information, or

hosted on a not-for-profit repository, e.g. FigShare, with access details provided in the manuscript.

- 'n' clearly defined (e.g. x cells from y slices in z animals) in the Methods. Authors should be mindful of pseudoreplication.

- All relevant 'n' values must be clearly stated in the main text, figures and tables.

- The most appropriate summary statistic (e.g. mean or median and standard deviation) must be used. Standard Error of the Mean (SEM) alone is not permitted.

- Exact p values must be stated. Authors must not use 'greater than' or 'less than'. Exact p values must be stated to three significant figures even when 'no statistical significance' is claimed.

All data points are plotted on the figures, although $n > 30$. Raw data cannot be made freely available due to privacy/ethical restrictions. However, data can be made available on request. The definition of "n" is clearly provided. Results are reported as mean \pm standard deviation [range]. P-values show three significant figures even when no statistical significance is being reported.

- Please include an Abstract Figure file, as well as the Figure Legend text within the main article file. The Abstract Figure is a piece of artwork designed to give readers an immediate understanding of the research and should summarise the main conclusions. If possible, the image should be easily 'readable' from left to right or top to bottom. It should show the physiological relevance of the manuscript so readers can assess the importance and content of its findings. Abstract Figures should not merely recapitulate other figures in the manuscript. Please try to keep the diagram as simple as possible and without superfluous information that may distract from the main conclusion(s). Abstract Figures must be provided by authors no later than the revised manuscript stage and should be uploaded as a separate file during online submission labelled as File Type 'Abstract Figure'. Please also ensure that you include the figure legend in the main article file. All Abstract Figures should be created using BioRender. Authors should use The Journal's premium BioRender account to export high-resolution images. Details on how to use and access the premium account are included as part of this email.

An abstract figure has been attached to the revised manuscript, and the caption added to the very end of the manuscript.

EDITOR COMMENTS

Reviewing Editor:

Thank you for submitting your manuscript to The Journal of Physiology. Two experts in the field have reviewed your work and I am pleased to report that both are enthusiastic about the merits of the paper and its potential influence for the wider field. Both referees have suggestions that should be carefully considered. In general these encourage the authors to view the data through different lenses to ensure the most comprehensive considerations are included in the final manuscript. I look forward to receiving your responses to the referees and a revised manuscript.

The team extends its sincere thanks to the reviewers and editors for their positive feedback, constructive comments, and valuable suggestions, which have greatly enhanced the clarity and

quality of this manuscript. We have addressed each point raised, and the manuscript has been revised accordingly, as detailed below.

Senior Editor:

Thank you for submission of your research article to the Journal of Physiology for consideration. The article has been reviewed by experts in the field and found to be potentially acceptable for publication pending adequate revision to address all of the concerns raised. Please address all comments from the external referees and reviewing editor as well as addressing the list of requirements or publication in the journal included in this letter and the link below.

<https://physoc.onlinelibrary.wiley.com/pb-assets/hub-assets/physoc/documents/TJP-Rigour-and-Reproducibility-Requirements-1724673661727.pdf>

Thank you. All comments have been addressed, and the publication requirements have been met.

REFEREE COMMENTS

Referee #1:

This study builds on several prior publications from the same research group, utilising a methodology in which they have considerable expertise to map the distribution of neural drive to the genioglossus (GG) in individuals with and without obstructive sleep apnea (OSA). The authors found no significant differences in electromyographic (EMG) activity between the different compartments of the GG. Likewise, no differences were observed between OSA patients and healthy subjects.

The manuscript is well-written and makes a significant contribution to the existing body of knowledge on the neural control of the GG. The methodology is thoroughly described, and despite the complexity of the results, they are presented in a clear and comprehensible manner. The authors have also appropriately discussed the important limitations of the study. I have only minor comments and suggestions for consideration:

1. Electrodes placement: According to Figure 1, the posterior-segment electrodes were placed relatively close to the anterior ones, effectively in the middle of the GG. It is possible that placing these electrodes further posteriorly might reveal larger differences between the anterior and posterior compartments.

We agree that positioning the electrodes further posteriorly could potentially reveal larger differences between the anterior and posterior compartments. However, our setup was designed to balance precision and safety while aligning with previous studies (e.g. Yeung et al., 2022). As a result, the electrodes were placed approximately 1 cm apart to target both the anterior and posterior compartments of the genioglossus (GG).

This placement was selected to ensure consistent and reliable data acquisition while minimising the potential risks associated with more posterior insertions, such as inaccurate placement (i.e. ultrasound imaging is less effective for guiding precise electrode placement in the more posterior

regions of the GG, particularly in those with large tongues for which it can be difficult to visualise the whole tongue volume), or complications arising from more posterior positioning (i.e. electrodes positioned further posteriorly may also be less stable due to increased muscle movement in the oropharynx during swallowing or breathing).

Future studies could certainly explore alternative electrode placements to further investigate this hypothesis in greater detail. This limitation was added to the manuscript, page 22.

"Fifth, positioning the posterior electrodes further back could have revealed larger differences between the anterior and posterior compartments. However, our setup followed similar multielectrode recording approaches used by us (Yeung et al., 2022)(Carberry et al., 2022) and others (Eastwood P, et al., 2003). This placement ensured reliable data while minimising risks, such as less precise ultrasound-guided placement in more posterior regions (especially in larger tongues, for which the back of the tongue might not be seen) and reduced electrode stability due to increased oropharyngeal movement. Future studies could explore alternative placements to further investigate this hypothesis".

2. Table 1: Why were the OSA patients hyperpneic, with high tidal volumes? This ventilation pattern could influence EMGgg activity, which is often linked to diaphragm activity, and also affect Pepi. Could this observation impact the comparison between participants with and without OSA, and might it warrant statistical correction?

On average, the OSA group exhibited a larger tidal volume than controls, which may be partially due to their higher BMI (mean difference: 2.7 kg/m²). While this BMI difference was not statistically significant (p = 0.065), it could still have influenced breathing patterns.

We recognise that ventilation patterns, including tidal volume, may influence genioglossus EMG activity due to its association with diaphragm activity and nadir Pepi. While we did not directly measure diaphragm activity, we accounted for its potential influence by including nadir Pepi as a covariate in our statistical models. This approach aimed to minimise any confounding effects on group comparisons.

To acknowledge this consideration, we have revised the final limitation of the manuscript (page 22) as follows:

*"Lastly, participants with OSA had a mean BMI of almost 3 kg/m² higher than the controls but still within the non-obese range. Although no statistically significant difference between groups was observed, this may have influenced comparisons between groups, **as the OSA group breathed with a larger tidal volume than the control group, and ventilation patterns may influence genioglossus activity. By including nadir Pepi as a covariate in our models, we aimed to control for this effect and reduce its potential impact on group comparisons.**"*

3. OSA severity range: The authors should provide the number of OSA patients in each severity range (mild, moderate, severe). If the sample includes a substantial proportion of patients with mild OSA, the authors should consider comparing normal controls to patients with moderate and severe OSA only (or apply an alternative statistical approach) to evaluate a potential relationship between EMG activity and OSA severity.

Thank you for the suggestion. Our cohort includes 16 participants with mild OSA ($5 < \text{AHI} \leq 15$ events/hr), 14 with moderate OSA ($15 < \text{AHI} \leq 30$ events/hr), and 15 with severe OSA ($\text{AHI} > 30$ events/hr). This is now included with the other participant characteristics on page 11.

We have also revised the statistical analysis in the manuscript to investigate the effect of OSA severity (AHI) instead of OSA status, showing:

- Table 3: There is no effect of AHI on the odds of a compartment having a phasic or tonic activity pattern,
- Table 6: Across all compartments with a phasic pattern, there was no effect of AHI on the Log10-transformed EMG activity,
- Table 7 and Figure 7: In the horizontal posterior compartment with phasic activity pattern, a higher AHI was associated with a lower Log10-transformed phasic EMG, although the effect was small (Beta = -0.007, 95% CI [-0.012 – -0.002], $t(100) = -2.820$, $P = 0.006$).

4. Figure 4: The lack of statistically significant differences might be partly due to the statistical analysis used, which integrated all data. For instance, when examining the regional inspiratory activity pattern of the anterior oblique compartment in OSA and non-OSA groups, the difference appears significant despite the small number of controls. Please comment on this observation.

The second reviewer also noted this issue (comment #2). As such, the statistical analysis was extensively revised based on the various comments we received by running additional mixed linear models with and without the inclusion of the OSA severity effect. The key new findings of the revised manuscript are:

- There was a compartmental variation (oblique-horizontal) in EMG activity when adjusted for nadir Pepi (Table 4 and Figure 5).
- Higher phasic activity correlates with a more negative nadir Pepi only in the oblique compartments (Table 5 and Figure 6).
- OSA did not have an additive effect on top of the more negative nadir Pepi on EMG activity (Table 6).

All these results were observed in genioglossus compartments with a phasic activity pattern.

5. Awake state recordings: The authors recognise that limiting EMG recordings to the awake state is a limitation, as the neural control of the GG changes significantly during sleep. However, they should also clarify the relevance of comparing awake GG activity during relaxed breathing in subjects with and without OSA.

Thank you for highlighting this important point. There has been some discussion in the literature about whether there may be changes in upper airway mechanics and control in OSA that extend to wakefulness. E.g. if, as has been suggested, there are OSA-related remodelling or injury to the innervating nerves, this would be expected to be discernable during wakefulness as well as sleep. Also, if differences in upper airway function can be observed awake, there is potential for clinical use, although we have not explored that here.

This point has now been made clear in the related limitation page, page 22:

"Second, ~~only~~ awake data were collected, as measuring GG activity during the awake state provides valuable insights into the baseline neuromuscular properties and compensatory mechanisms that may differ between individuals with and without OSA, including those that are not sleep-

dependent. However, it is known that the sleep state modifies central output to the genioglossus and attenuates upper airway reflexes (Sauerland & Harper, 1976; Cori et al., 2018; Gell et al., 2022). Therefore, further work is needed to investigate compartmental drive during different sleep stages."

It was further emphasised at the end of the introduction, page 6:

"Awake-state recordings are important for understanding the predisposition to airway collapse during sleep and may help to identify key neuromuscular features that differentiate OSA and non-OSA populations."

6. Pre-inspiratory inspiratory activity: As noted by the authors, the pre-inspiratory inspiratory activity of upper airway dilators and its significance has been recognised for more than 40 years. Nevertheless, the role of this central "preparatory" drive in OSA pathophysiology has not been validated. Since this topic is beyond the scope of the current study, the discussion on this subject could be abbreviated. Conversely, the authors might consider suggesting potential implications of their findings for recent treatment modalities, such as hypoglossal stimulation and myofunctional therapy.

We agree and have amended this section of the discussion (last paragraph, page 21) to highlight that *"the role of this central preparatory drive in OSA pathophysiology is not well understood"*, and that *"Understanding of the role of central drive in maintaining airway patency could inform novel interventions designed at enhancing preparatory muscle activation, such as hypoglossal nerve stimulation or targeted myofunctional therapy, to improve airway stability during sleep"*.

Referee #2:

Deficits in genioglossus reflex activity are key to the development of obstructive sleep apnea, but detailed physiology studies are lacking. In the current study, Jugé and colleagues studied a large number of OSA patients and healthy controls during quiet breathing to assess whether 1), there are differences in genioglossus EMG between oblique and horizontal compartments and 2), whether compartment-specific genioglossus EMG activation is different in OSA versus controls. As part of the rationale, the authors expected that EMG would be greater in the horizontal vs oblique compartments, and that EMG activation would be greater in OSA versus controls (compensation for the greater negative pharyngeal pressure in OSA). In analyses adjusting for the epiglottic pressure swings, the authors found that there were no differences in genioglossus EMG levels between oblique and horizontal compartments. The authors also found no adjusted differences between OSA and controls in genioglossus EMG levels.

I have several recommendations to improve the presentation and focus on the primary aims.

1) To address their Aim 1, the authors provided figures showing raw sample data with means/SD (Figure 5, then adjusted mixed model F-values and P-values (Table 4), but did not provide adjusted EMG (least-squares) differences (and 95%CI) across compartments. Providing beta values for each of the compartments (there should be three difference terms) is needed to interpret the similarities in genioglossus EMG; likewise, 95%CI are needed to interpret the stability of the estimates.

Table 4 has been amended to include beta values and 95%CI.

2) Based on the figures, it seems like there may be a quantitatively meaningful increase in peak

activity in oblique compared to horizontal (particularly horizontal). I encourage the authors to more carefully present the results relating to their primary hypothesis. The primary model presented in Table 4 may be over interpreted: The model describes a multi-level term for compartment, OSA status, and a compartment-by-OSA status interaction term. The inclusion of the interaction term, therefore, means that the main effect for compartment, as presented, is based on data primarily from controls (which is considerably more limited in sample). I suggest first presenting results without inclusion of the OSA status term.

Thank you for the suggestion. To address Aim 1 (compartmental differences in EMG measurements), the effect of genioglossus compartments alone (i.e., without the main effect of OSA status and their interaction) on Log10-transformed quantitative EMG measurements is now presented in Table 4. We followed this analysis with a Bonferroni-adjusted pairwise comparison to further characterise compartmental differences in EMG activity (Figure 5). We found that Log10-transformed peak, phasic, and tonic EMG were higher in the oblique than in the horizontal compartments across all compartments with a phasic activity pattern. See response to comment#6 below for why we log-transformed data.

3) The rationale for removing phasic components from an analysis of the strength of phasic EMG activity is not entirely clear. For example, the horizontal compartments appear to be less-often phasic (Figure 4) and the phasic components also appear to have less activity; if the authors instead chose to combine tonic and phasic compartments (in what I would see as a less biased analysis) then they may have found smaller EMG activity on average in the horizontal vs oblique compartments. I suggest not parsing out compartments by tonic/phasic status, at least in the primary analyses.

If we understand the comment correctly, the reviewer is querying whether we should have removed the tonic compartments from the quantitative analysis of the amplitude of the phasic EMG. We acknowledge the importance of considering GG compartments with both tonic and phasic EMG pattern in EMG analysis (as we have done in Figure 4 and Table 3). However, since we did not find any differences in the distribution of tonic vs phasic activity across compartments (Figure 4 and accompanying analysis, page 13), we chose to analyse the phasic components separately. This decision was made because tonic compartments are unlikely to be contributing to dynamic mechanisms to oppose the negative airway pressures occurring during inspiration.

This has now been clarified in the manuscript, page 10:

"In this section, only recordings with a phasic activity pattern were analysed since tonic EMG patterns are unlikely to be contributing to dynamic mechanisms to oppose the negative airway pressures occurring during inspiration"

Importantly also, the peak EMG values reported in our quantitative analysis incorporate both tonic and phasic contributions in GG compartments exhibiting phasic activity, ensuring that our analysis captures the overall strength of EMG activation.

We appreciate the reviewer's perspective and are happy to clarify further if needed.

4) In the assessment of OSA versus controls, I suggest performing analysis without and then with Pepi adjustment: It would be important to know if EMG in OSA is > EMG in controls for comparison to the prior Mezzinotte/Malhotra papers in the literature, then to see OSA-vs-control differences disappear with Pepi adjustment. The rationale also alludes to Pepi differences as being a reason for OSA related increase in GG activity so adjusting Pepi effects out in primary analysis appears unhelpful.

We conducted the analysis without adjusting for nadir PePi and found that OSA severity did not significantly influence the odds of a GG compartment exhibiting a phasic activity pattern ($P = 0.999$), as the interaction between AHI and GG compartment ($P = 0.956$). The full statistical output of the binary logistic regression is provided below:

Variables in the Equation

		B	S.E.	Wald	df	Sig.	Exp(B)	95% C.I. for EXP(B)	
								Lower	Upper
Step 1 ^a	AHI	.000	.017	.000	1	.999	1.000	.967	1.034
	GG compartment			2.392	3	.495			
	Horizontal Anterior	-.075	.683	.012	1	.913	.928	.243	3.540
	Horizontal Posterior	-.640	.694	.850	1	.356	.527	.135	2.054
	Oblique Anterior	-.946	.791	1.433	1	.231	.388	.082	1.828
	AHI * GG compartment			.319	3	.956			
	AHI by Horizontal Anterior	-.002	.021	.006	1	.939	.998	.958	1.041
	AHI by Horizontal Posterior	-.003	.022	.023	1	.880	.997	.954	1.041
	AHI by Oblique Anterior	.009	.025	.138	1	.710	1.009	.960	1.061
	Constant	.049	.547	.008	1	.928	1.050		

a. Variable(s) entered on step 1: AHI, GG compartment, AHI * GG compartment.

Furthermore, this non-significant effect of OSA severity on GG activation was also observed in the log₁₀-transformed peak, phasic, tonic, and inspiratory EMG analyses (see the next 4 mixed linear model outputs).

Estimates of Fixed Effects^a

Parameter	Estimate	Std. Error	df	t	Sig.	95% Confidence Interval	
						Lower Bound	Upper Bound
Intercept	.766	.050	104	15.263	<.001	.667	.866
AHI	-.001	.002	104	-.454	.651	-.004	.003

a. Dependent Variable: log₁₀_peakEMG.

Estimates of Fixed Effects^a

Parameter	Estimate	Std. Error	df	t	Sig.	95% Confidence Interval	
						Lower Bound	Upper Bound
Intercept	.521	.058	104	8.968	<.001	.406	.636
AHI	-.001	.002	104	-.307	.760	-.004	.003

a. Dependent Variable: log₁₀_phasicEMG.

Estimates of Fixed Effects^a

Parameter	Estimate	Std. Error	df	t	Sig.	95% Confidence Interval	
						Lower Bound	Upper Bound
Intercept	.380	.044	104	8.668	<.001	.293	.467
AHI	-.001	.001	104	-.641	.523	-.004	.002

a. Dependent Variable: log10_tonicEMG.

Estimates of Fixed Effects^a

Parameter	Estimate	Std. Error	df	t	Sig.	95% Confidence Interval	
						Lower Bound	Upper Bound
Intercept	.565	.055	104	10.243	<.001	.456	.674
AHI	-.003	.002	104	-1.404	.163	-.006	.001

a. Dependent Variable: log10_InspEMG.

While this suggests that OSA severity alone does not significantly influence GG activity, it is important to consider, as noted by the first reviewer (comment #2), that our two groups of participants exhibited different ventilation patterns, which could influence genioglossus activity. Therefore, to account for this potential confounding factor and better assess the effect of OSA severity (or OSA status) on genioglossus activity, we included nadir Pepi as a covariate in our analysis. This approach allows us to control for the impact of breathing patterns and airway pressure on genioglossus activity and isolate the specific effect of OSA. This adjustment is now made clear in the manuscript on page 9 (statistical analysis).

“To control for the impact of airway pressure on genioglossus multiunit activity, nadir Pepi was included as a covariate in analyses”.

We hope this explanation addresses the reviewer’s concern and demonstrates that the inclusion of nadir Pepi as a covariate was a necessary step to ensure a rigorous and unbiased analysis of OSA-related differences in GG activity.

5) I suggest considering performing an exploratory analysis that uses separate terms for anterior/posterior status, and horizontal/oblique status (replacing the 4-level term). This analysis may have more statistical power due to fewer terms, and also appears more consistent with the introductory rationale. Note that the anterior/posterior status appears less important than horizontal/oblique status.

Thank you for this thoughtful suggestion. We agree that performing an exploratory analysis using separate terms for anterior/posterior status and horizontal/oblique status, rather than the current four-level term, could provide greater statistical power. However, as presented in our introduction, previous literature suggested an anterior-posterior variation in genioglossus activation (Eastwood et al. 2003; Vranish & Bailey, 2015). As such, when investigating the oblique-horizontal variation, not including anterior-posterior measurements might have overlooked potential regional variation in EMG activities. Given this, we have opted to retain the four-level term in our analysis.

6) EMG results appear not-normally distributed, you may like to square-root transform results before analysis.

Thank you for this excellent suggestion. We agree that transforming the EMG data improves its distribution for analysis. In collaboration with our biostatistician, we considered a square-root transformation, but we found that a Log10 transformation provided a better approximation of normality. This was determined through visual inspection of Q-Q plots of the residuals from the mixed linear models and scatter plots of residuals versus predicted values (see examples below).

7) The authors used %max for analysis, but the max maneuvers may have activated some compartments more than others. Have the authors considered using data in uV units, or considered adjusting for the %max calibration uV levels.

We agree that this might be a concern. However, our analysis showed that the maximum EMG values from either tongue protrusions or swallows did not differ significantly between compartments (mixed linear regression, $F(3,180) = 1.488$, $P = 0.219$). This is indicated on page 12 of the manuscript. Therefore, we consider our normalisation method adequate, and no changes have been made.

8) To understand if "genioglossus responsiveness" depends on compartment type, the authors may like to see include a compartment-by-Pepi term to test whether compartments differ in their response to negative pressure (suggest not including OSA status in this analysis due to unnecessary complexity).

We conducted the suggested analysis to examine the interaction effects of the genioglossus compartment and nadir Pepi on Log10-transformed phasic EMG. Our findings indicate that only the anterior and posterior oblique compartments showed a significant interaction (see updated Table 5 and revised Figure 6B). In contrast, the interaction effects for the horizontal compartments were not significant (see revised Figure 6C).

Minor:

-Beta values are missing units, are these %max/cmH2O or uV/cmH2O?

Log10 (%max)/cm H₂O. Units have been added where required.

Table 4 illustrates a multivariable regression but shows only F and P values rather than betas and 95%CI, which makes it difficult to interpret if there are any physiologically-meaningful differences present that may have not reached significance. Units are missing (%max?). e.g. There are four GG compartments, yet we can't see the beta for each of them, only the F value for all of them combined.

Table 4 has been amended to report beta values and 95%CI of the model. Results for all genioglossus compartments have also been included.

Figure 6. too many decimal points in the values within the figures.

The number of decimals has been reduced in revised Figure 6.

Re: ". These results reject our hypotheses."

-Technically your results did not support your hypotheses, please rephrase.

The sentence has been amended accordingly, page 17: "These results **do not support** our hypotheses".

END OF COMMENTS

Dear Professor Bilston,

Re: JP-RP-2025-287943R1 "Compartmental inspiratory genioglossus electromyographic activity in supine, awake individuals with and without obstructive sleep apnoea" by Lauriane Jugé, Peter GR Burke, Jade Yeung, Fiona L Knapman, Elizabeth C. Brown, Alan KI Chiang, Danny J Eckert, Jane E Butler, and Lynne E Bilston

We are pleased to tell you that your paper has been accepted for publication in The Journal of Physiology.

Yours sincerely,

Harold Schultz
Senior Editor
The Journal of Physiology

If you would like to receive our 'Research Roundup', a monthly newsletter highlighting the cutting-edge research published in The Physiological Society's family of journals (The Journal of Physiology, Experimental Physiology, Physiological Reports, The Journal of Nutritional Physiology and The Journal of Precision Medicine: Health and Disease), please click this link, fill in your name and email address and select 'Research Roundup':
<https://www.physoc.org/journals-and-media/membernews>

- You can help your research get the attention it deserves! Check out Wiley's free Promotion Guide for best-practice recommendations for promoting your work at: www.wileyauthors.com/eoo/guide. You can learn more about Wiley Editing Services which offers professional video, design, and writing services to create shareable video abstracts, infographics, conference posters, lay summaries, and research news stories for your research at: www.wileyauthors.com/eoo/promotion.

EDITOR COMMENTS

Reviewing Editor:

Comments to the Author:

Thank you for your careful and thorough response to the referees' comments and for making revisions to the text. All points

have been satisfactorily addressed. Thank you for this important contribution to The Journal further highlighting complexity in the neuromechanical control of the tongue in human OSA, which should prove highly influential to the field.

Senior Editor:

Comments to the Author:

The editors thank the authors for these final adjustments to the manuscript. The article is now accepted for publication. Congratulations on an interesting and insightful study. Please consider the Journal of Physiology for your future studies.

REFeree COMMENTS

Referee #1:

The authors answered my comments fully and satisfactorily. I have no further comments.

Referee #2:

The authors have taken my recommendations on board to improve the manuscript. No further recommendations.